# Molecular and mechanical signatures contributing to mouse epidermal differentiation and barrier formation

Alexandra Prado-Mantilla[1,2], Wenxiu Ning[1,2†], Terry Lechler[1,2]*

[1]Department of Dermatology, Duke University Medical Center, Durham, United States; [2]Department of Cell Biology, Duke University Medical Center, Durham, United States

## eLife Assessment

The authors address a **fundamental** question for cell and tissue biology. They use the skin epidermis as a paradigm and ask how stratifying self-renewing epithelia induce differentiation and upward migration in basal dividing progenitor cells to generate suprabasal barrier-forming cells that are essential for a functional barrier formed by such an epithelium. The authors provide **compelling** evidence time that an increase in intracellular actomyosin contractility, a hallmark of barrier-forming keratinocytes, is sufficient to trigger terminal differentiation, providing in vivo evidence of the interdependency of cell mechanics and differentiation. To illustrate their points, the authors use a combination of genetic mouse models, RNA sequencing, and immunofluorescence analysis. Precisely how the changes in gene expression, cell morphology, mechanics, and cell position are instructive and whether consecutive changes in differentiation are required still remain unclear, but the paper takes a nice step in advancing our knowledge of the process.

**Abstract** Formation of the skin barrier requires rapid proliferation coupled with differentiation and stratification of the embryonic epidermis. Basal progenitors give rise to progeny throughout development – first to intermediate cells, a transient proliferative suprabasal cell population, and later to spinous cells. Neither the function nor the differentiation trajectory of intermediate cells has been documented. We generated transcriptomes of intermediate and spinous cells and identified specific markers that distinguish these two populations. Further, we found that intermediate cells express a subset of genes in common with granular cells of the epidermis – the terminal living cell type that helps establish the barrier. Lineage tracing revealed that most intermediate cells directly transition to granular cells without expressing markers specific to spinous cells, thus revealing a distinct lineage pathway leading to granular fate. In addition to their transcriptional similarities, intermediate and granular cells both had hallmarks of increased actomyosin contractility. We found that rather than simply lying downstream of cell fate pathways, contractility was sufficient to suppress spinous fate and promote granular gene expression. Together, these data establish the molecular and mechanical characteristics of the developing epidermis that allow this tissue to rapidly develop barrier activity.

*For correspondence:
Terry.Lechler@duke.edu

Present address: †Center for Life Sciences, School of Life Sciences, Yunnan Key Laboratory of Cell Metabolism and Diseases, Yunnan University, Kunming, China

**Competing interest:** The authors declare that no competing interests exist.

## Introduction

Development of many tissues is characterized by rapid proliferation coupled with morphogenesis and differentiation. This process often uses specialized and transient cell types that are not found in homeostatic tissue, necessitating distinct pathways for differentiation (*Singh and Tiwari, 2023*).

During the embryonic development of the epidermis, an initial single layer of progenitor cells gives rise to a multilayered and differentiated tissue that acts as a chemical and mechanical barrier at birth (*Moreci and Lechler, 2020*; *Sumigray and Lechler, 2015*). This process begins at embryonic day (E) 14.5 in the mouse back skin when basal progenitor cells start to give rise to a suprabasal cell layer of intermediate cells (ICs) (*Damen et al., 2021*; *Lechler and Fuchs, 2005*; *Smart, 1970*). Subsequently, basal cells divide to generate spinous cells, which are postmitotic. These cells then further differentiate into granular cells, the last living cell types in the epidermis which are integral to forming the epidermal barrier. ICs are a transient cell type that expresses differentiation markers like keratin 1 (K1) and K10 but remain mitotically active (*Weiss and Zelickson, 1975*). At this time point, the proliferation rate of ICs is comparable to basal cells, suggesting that they are a significant contributor to tissue expansion (*Damen et al., 2021*; *Smart, 1970*). These cells lack basement membrane attachment and, thus, they proliferate in a tissue environment that is normally quiescent in the adult. Currently, we do not understand if they play functional roles outside of proliferation, their cell fate trajectory, or how they compare to later generated spinous cells at a molecular level. Further, we lack specific markers for ICs despite recent scRNA-Seq data that has characterized the transcriptomes of these cells (*Jacob et al., 2023*). Generation of ICs is transient (E14–15.5), and by E16.5 basal cells give rise to postmitotic spinous cells (*Figure 1A*). It is not known whether this switch in progeny type is driven by intrinsic changes in basal progenitors, changes in signals from the surrounding periderm, and/or by systemic signals (circulating factors or amniotic fluid; *Huebner et al., 2012*).

In addition to chemical cues, mechanical information is assessed by cells and can instruct proliferation and differentiation decisions (*Vining and Mooney, 2017*). This has been most clearly demonstrated by substrates of differential stiffness eliciting altered differentiation pathways in mesenchymal stem cells (*Engler et al., 2006*). Whether intra-tissue mechanical information is also instructive in differentiation has not been as thoroughly addressed. That said, in many tissues, differentiation leads to changes in contractility and/or stiffness. For example, in the epidermis, there is increased contractility in the granular cell layer, and this contractility is important for the formation of tight junctions in this cell layer (*Miroshnikova et al., 2018*; *Rübsam et al., 2017*; *Sumigray et al., 2012*). While this flow of information from transcriptome to contractility has largely been assumed to be unidirectional, some works have demonstrated bidirectional interactions (*Le et al., 2016*; *Meyer-ter-Vehn et al., 2006*). When physical linkages between the cytoskeleton and nucleus were removed, there was a premature differentiation of keratinocytes, and this was hypothesized to involve alterations in the transcription of the epidermal differentiation complex (EDC), a genetic locus that undergoes a change in localization in response to differentiation (*Carley et al., 2021*). Further, increased contractility in differentiated cells of the epidermis results in a non-cell autonomous effect on proliferation of their progenitors (*Ning et al., 2021*). However, we lack clear evidence that contractility can directly affect differentiation in the epidermis.

Here, we used bulk sequencing to characterize the transcriptomes of ICs and spinous cells, as well as their progenitors. We defined markers that are specific for ICs and spinous cells and showed marked transcriptomic differences in the basal cell progenitors that give rise to these distinct cell types. Further, our data demonstrate that ICs express many genes associated with granular cells – a cell type that emerges days later and is responsible for secreting lipids that constitute part of the epidermal barrier. Rather than moving through a spinous state, as has long been assumed (*Koster and Roop, 2005*), ICs appear to directly transition to granular cells. ICs also show similarity to granular cells in having increased apparent actomyosin contractility. Using genetic models to induce contractility in spinous cells, we demonstrate that contractility is sufficient to induce a granular-like state. Together, this work reveals the lineage pathways and the influences of contractility on epidermal barrier formation.

## Results

### Transcriptomic analysis of embryonic epidermal differentiation

During early epidermal stratification at E14.5, ICs are the first layer of suprabasal cells that express the differentiation marker K10 and proliferate at a rate similar to basal cells (*Figure 1A–C*; *Damen et al., 2021*; *Smart, 1970*). The proliferation rate of suprabasal cells decreases over the next 48 hr, and these cells are largely postmitotic by E16.5. To further validate these suprabasal cell dynamics during

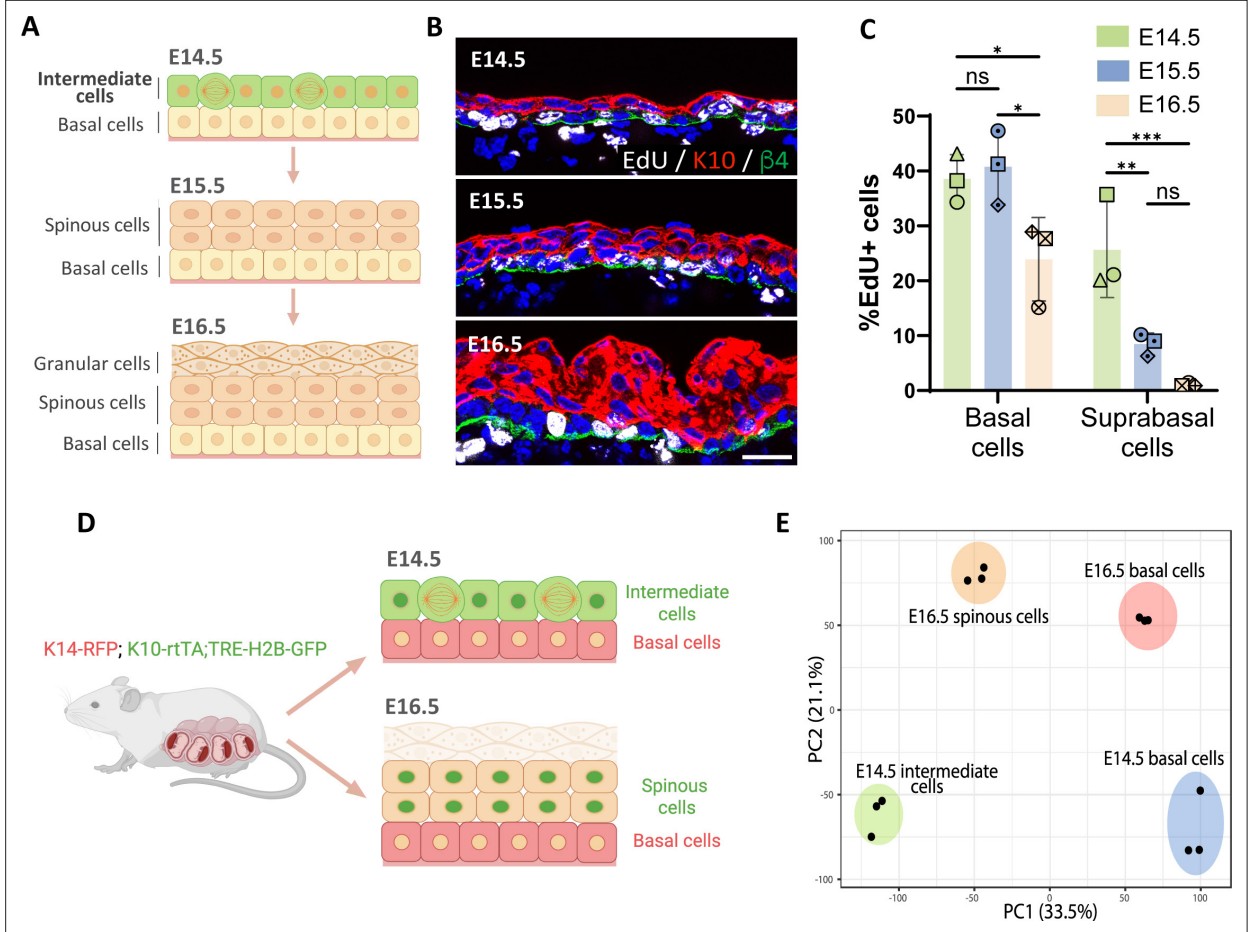

**Figure 1.** Transcriptomic analysis of embryonic epidermal cell populations. (**A**) Diagram depicting the stratification process and different cell populations (basal and K10+ suprabasal layers) existing in the epidermis from embryonic day (E) 14.5 to E16.5. Created with BioRender.com. (**B**) Images of EdU staining (white) at E14.5, E15.5, and E16.5. Immunofluorescence of suprabasal cells labeled with keratin 10 (K10) in red and the basement membrane, with β4-integrin in green. Scale bars: 20 μm. (**C**) Quantification of the percentage of EdU+ cells in the basal and the first two layers of suprabasal cells at E14.5, E15.5, and E16.5. n=3 embryos/time point. Different shapes represent different embryos. Data are presented as mean ± standard deviation (SD). Ordinary two-way ANOVA (p=0.0003), Tukey's multiple comparisons test, ns: not significant, *: p<0.05, **: p<0.01, ***: p<0.001. (**D**) Schematic showing basal and suprabasal cell populations collected for RNA sequencing: K14-RFP;K10-rtTA;TRE-H2B-GFP pregnant dams were all fed with doxycycline from E12.5 and then sacrificed at either E14.5 or E16.5. The E14.5 intermediate cells (K10-GFP⁺;K14-RFP⁻), E16.5 spinous cells (K10-GFP⁺;K14-RFP⁻), and basal cells at E14.5 and E16.5 (K10-GFP⁻;K14-RFP⁺) cells were collected separately and sent for RNA sequencing. Granular cells were excluded from E16.5 samples. Created with BioRender.com. (**E**) Principal component analysis (PCA) score plot of the first two principal components (PC1: 33.5% variance and PC2: 21.1% variance) for gene expression levels from samples of cell populations indicated in (**D**) (n=3 embryos/ cell population).

The online version of this article includes the following figure supplement(s) for figure 1:

**Figure supplement 1.** Live imaging of mitotic suprabasal cells during early epidermal stratification.

epidermal stratification, we performed live imaging of back skin explants from K10-rtTA;TRE-H2B-GFP embryos, which express H2B-GFP in K10-positive suprabasal cells. Consistent with data from fixed embryos, we observed numerous suprabasal mitotic events in explants collected from E14.5 embryos, while suprabasal cells in explants from E16.5 embryos were mitotically inactive (*Figure 1—figure supplement 1*). All divisions visualized were planar to the epithelium.

To understand the differences between suprabasal cells at E14.5 and E16.5, we explored the transcriptomic changes occurring between these two time points. We used K14-RFP;K10-rtTA;TRE-H2B-GFP mice and FACS to purify H2B-GFP⁺ suprabasal cells and RFP⁺ basal cells at E14.5 and E16.5 and performed bulk RNA-Seq of each of these cell populations (*Figure 1D*). Given that granular cells also form part of the suprabasal compartment at E16.5, we excluded the more superficial cells that

bind *Ulex europaeus* agglutinin I lectin from E16.5 samples in order to uniquely compare suprabasal cells that lie immediately above the basal layer (ICs and spinous cells) (*Brabec et al., 1980*).

Principal component analysis (PCA) of the sequenced cell populations revealed that biological replicates grouped together and that samples were segregated by cell type (basal vs suprabasal) in the first component, and by age in the second component (E14.5 vs E16.5). Notably, ICs at E14.5 and spinous cells at E16.5 were not clustered together, indicating that they are distinct populations, as are their corresponding basal cells (*Figure 1E*). Though not addressed further here, the developmental transition in suprabasal cells from E14.5 to E16.5 may reflect the very different states of basal cells at these time points.

To identify specific molecular markers for ICs and spinous cells, we filtered the genes that were uniquely enriched in ICs (at E14.5 compared to both E16.5 spinous cells and E14.5 basal cells) and in spinous cells (compared to both E14.5 ICs and E16.5 basal cells). The IC gene signature consisted of 610 genes and included *Scara5*, a scavenger receptor, St8sia6, which encodes a sialyltransferase enzyme, and *Tgm1* (transglutaminase 1), an enzyme involved in covalent cross-linking of proteins (*Figure 2A*, *Figure 2—figure supplement 1*). To validate these candidates, we performed RNAscope and detected mRNA of *Scara5* and *St8sia6* in ICs but not in suprabasal cells at E16.5 (*Figure 2B–E*). *Scara5* expression was specific to ICs, while *St8sia6* mRNA was also present in granular cells at E16.5 (*Figure 2C*). In contrast, St8sia6 protein was detected in the granular layers at E16.5 by antibody staining; however, it was not seen in ICs at E14.5 (*Figure 2—figure supplement 2*). This suggests that some granular genes are transcriptionally enriched in ICs but not yet appreciably expressed at the protein level. This included loricrin, whose expression was detected in ICs, but whose protein product was only found in granular cells. This could be due either to delayed production/build-up of the protein product or due to translational inhibition of these mRNAs. We also observed that another protein marker of granular cells, Tgm1, was also present in ICs, as described in more detail below (Figure 4C).

There were 163 genes whose transcripts were specifically upregulated in spinous cells at E16.5 (*Figure 2F*, *Figure 2—figure supplement 1*). Among this list, the transcription factor *MafB* was a prominent candidate since it has been implicated in epidermal differentiation in cultured human epidermal cells (*Lopez-Pajares et al., 2015*), and it is expressed in suprabasal cells in vivo during development (*Miyai et al., 2016*). Consistent with our transcriptional data, immunofluorescence staining of MafB at E16.5 revealed that it was present in the first suprabasal cell layers in the epidermis but was excluded from granular cells (*Figure 2 G,I*), and it was undetectable in ICs at E14.5. Another highly upregulated gene in this list, *Ptgs1*, which encodes the enzyme Cox1, was also enriched in spinous cells but not in ICs at the protein level (*Figure 2H and J*). Therefore, MafB and Cox1 are specific markers for spinous cells in the embryonic epidermis.

## MafB inhibits proliferation of ICs

To test the functional role of Maf transcription factors in determining spinous cell fate, we prematurely induced MafB expression in ICs. For this, we collected K10-rtTA;TRE-MafB-HA (hereafter called K10-MafB) embryos that were collected at E14.5, when suprabasal cells are ICs (*Figure 3—figure supplement 1A*). In these embryos, MafB-HA was expressed in about 30% of all suprabasal cells (*Figure 3—figure supplement 1B*). Examination of EdU incorporation revealed that HA+ cells had a significantly lower rate of incorporation than ICs in controls, demonstrating that MafB expression is sufficient to decrease cell proliferation (*Figure 3A and B*). Notably, however, we found that there was increased proliferation of surrounding MafB-negative cells in mutant embryos (*Figure 3A and B*), while the overall suprabasal proliferation rate was not significantly different in mutants vs controls (*Figure 3—figure supplement 1C*). This suggests a potential homeostatic mechanism for maintaining total proliferation during the epidermal stratification process. To determine whether MafB expression was sufficient to repress IC-specific gene expression, we examined IC markers, including Tgm1 and Scara5. While we found a decrease in the levels of Tgm1 protein, *Scara5* mRNA levels were unchanged (*Figure 3C–F*). These data demonstrate that MafB may repress parts of the IC signature but does not globally repress all markers. Given that it altered the mitotic status of ICs into a spinous-like state, we also examined whether MafB expression was sufficient for induction of spinous markers. However, we found that the spinous marker Cox1 was not induced by MafB (*Figure 3G and H*). Therefore, MafB is sufficient for some aspects of spinous fate (like loss of proliferation), but not for the expression of all spinous marker genes.

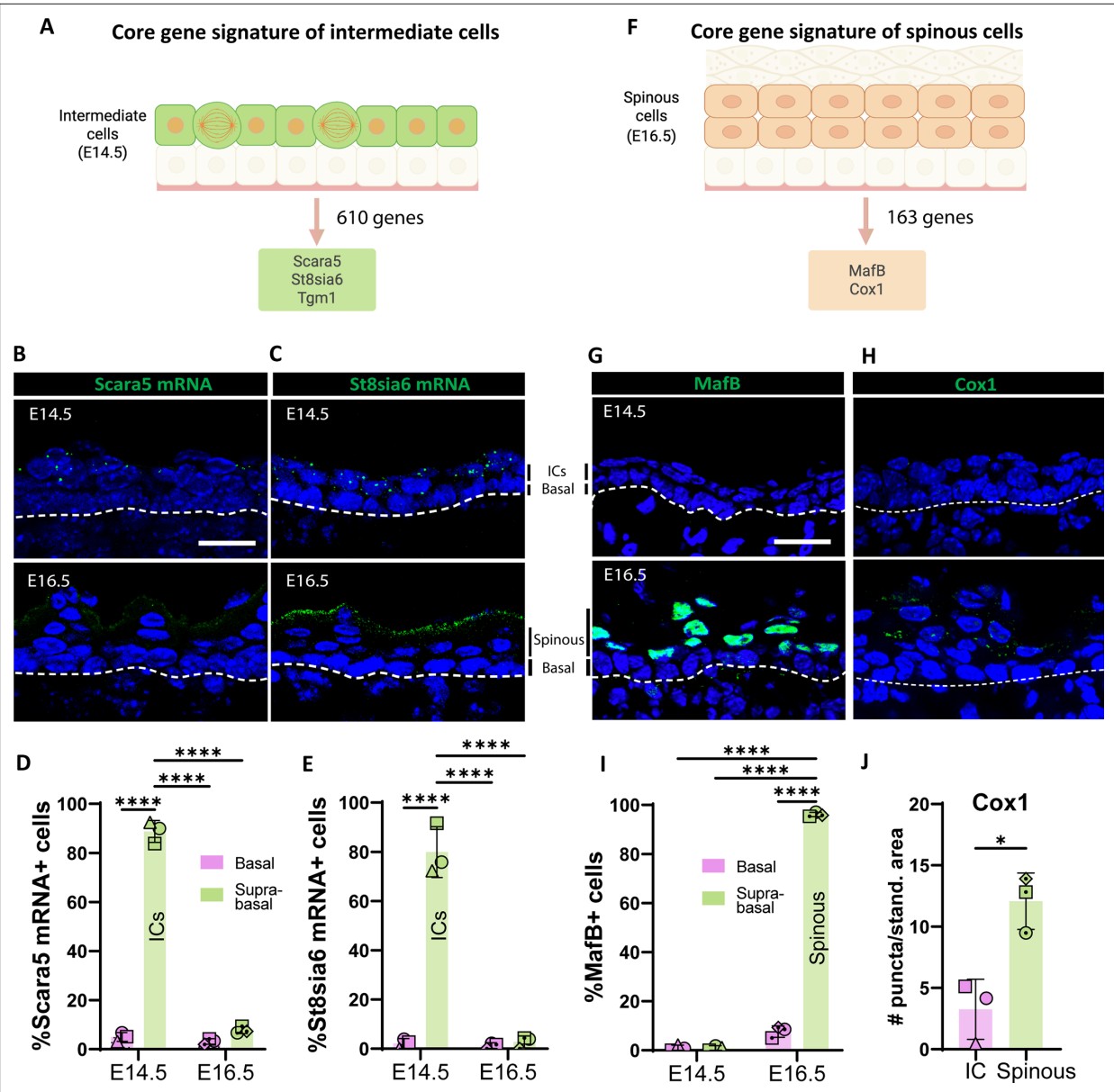

**Figure 2.** Identifying markers for intermediate (ICs) and spinous cells. (**A**) The core gene signature of ICs was obtained from the list of genes only upregulated in ICs (log2 fold change > 1, p<0.05; as compared with both basal cells at embryonic day [E] 14.5 and spinous cells at E16.5). Total genes in the IC signature were 610. From this list, Scara5 and St8sia6 were two of the most highly upregulated genes compared to spinous cells at E16.5. (**B** and **C**) RNAscope of Scara5 (**B**) and St8sia6 (**C**) in WT embryos at E14.5 and E16.5. Basement membrane is indicated as a dotted line. Scale bars: 20 µm. (**D** and **E**) Quantification of the percentage of Scara5+ (**D**) and St8sia6+ (**E**) basal and suprabasal cells at E14.5 and E16.5. n=3 embryos/time point. Different shapes represent different embryos. Data are presented as the mean ± SD. ****: p<0.0001, ordinary two-way ANOVA (p<0.0001), Sidak's multiple comparisons test. (**F**) The core gene signature of spinous cells was obtained from the list of genes only upregulated in spinous cells (log2 fold change > 1, p<0.05 compared with both basal cells at E16.5 and ICs at E14.5). Total genes in the spinous cell signature were 163. (**G** and **H**) Immunofluorescence staining of MafB (**G**) and Cox1 (**H**) in WT embryos at E14.5 and E16.5. Basement membrane is indicated as a dotted line. Scale bars: 20 µm. (**I**) Quantification of the percentage of MafB+ basal and suprabasal cells at E14.5 and E16.5. n=3 embryos/time point. Data are presented as the mean ± SD. ****: p<0.0001, ordinary two-way ANOVA (p<0.0001), Sidak's multiple comparisons test. (**J**) Quantification of Cox1 puncta/standardized area in ICs at E14.5 was compared to the first two layers of spinous cells at E16.5. n=3 embryos/time point. Data are presented as the mean ± SD. *: p<0.05, two-tailed unpaired t-test. Created with BioRender.com.

The online version of this article includes the following figure supplement(s) for figure 2:

**Figure supplement 1.** Genes upregulated in intermediate cells and spinous cells.

**Figure supplement 2.** Intermediate cells do not express some granular markers at the protein level.

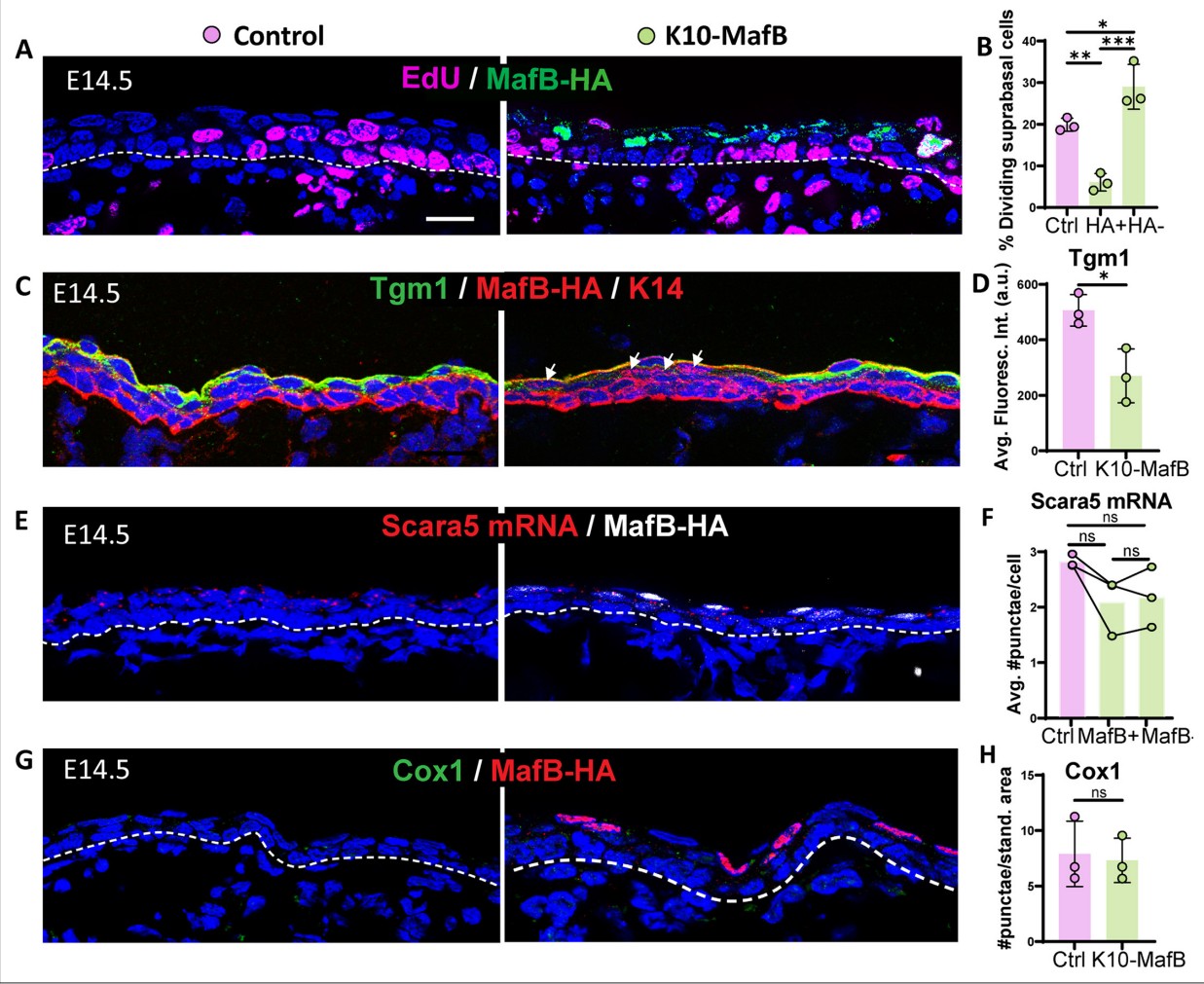

**Figure 3.** Mis-expression of MafB in intermediate cells (ICs) is sufficient to reduce proliferation but not to induce spinous fate. (**A**) Image of EdU staining (magenta) and MafB-expressing ICs immunolabeled with HA (green) in K10-MafB and control mice at embryonic day (E) 14.5, doxycycline fed since E12. All scale bars in this figure: 20 µm. Dotted lines represent the basement membrane. (**B**) Percentage of EdU+ dividing suprabasal cells in control (pink bars), and HA+ and HA– cells in K10-MafB mice (green bars) at E14.5 (n=3 embryos/genotype). Data are presented as the mean ± SD. *: p<0.05, **: p<0.01, ***: p<0.001, ordinary one-way ANOVA (p=0.0006), Tukey's multiple comparisons test. (**C**) Immunofluorescence staining of Tgm1 in K10-MafB vs control mice at E14.5. Arrows indicate HA+ cells in K10-MafB embryo. (**D**) Quantification of Tgm1 average fluorescence in suprabasal cells of control vs K10-MafB at E14.5 (n=3 embryos/genotype). Data are presented as the mean ± SD. *: p<0.05, two-tailed unpaired t-test. (**E**) RNAscope of Scara5 in red and immunostaining with MafB in white. (**F**) Quantification of the average number of Scara5 RNAscope puncta per cell in suprabasal layers of control, and MafB+ and MafB- cells of K10-MafB mice at E14.5 (n=3 embryos/genotype). Data are presented as the mean. ns: not significant, Friedman test (p=0.194), Dunn's multiple comparisons test. (**G**) Immunofluorescence staining of Cox1 in K10-MafB vs control mice at E14.5. (**H**) Quantification of number of Cox1 puncta per standardized area of suprabasal layer in K10-MafB vs control mice at E14.5. n=3 embryos/genotype. Data are presented as the mean. ns: not significant, two-tailed unpaired t-test.

The online version of this article includes the following figure supplement(s) for figure 3:

**Figure supplement 1.** The overall rate of suprabasal cell proliferation in K10-MafB is not significantly different from controls.

## ICs are granular cell precursors

Using the IC-specific gene signature we identified, we turned to Gene Ontology (GO) analysis to determine the biological pathway characteristics of these cells. Categories that were uniquely enriched in ICs at E14.5 included pathways involved in lipid metabolic processes and establishment of the skin barrier (*Figure 4A*). These pathways are hallmarks of granular cells, which produce and secrete lipids that contribute to the formation of the watertight cornified layer that acts as the epidermal barrier. These findings were unexpected since granular cells have not formed yet at E14.5. Comparing the IC gene signature to a granular gene signature that we generated from published data (*Matsui et al.,*

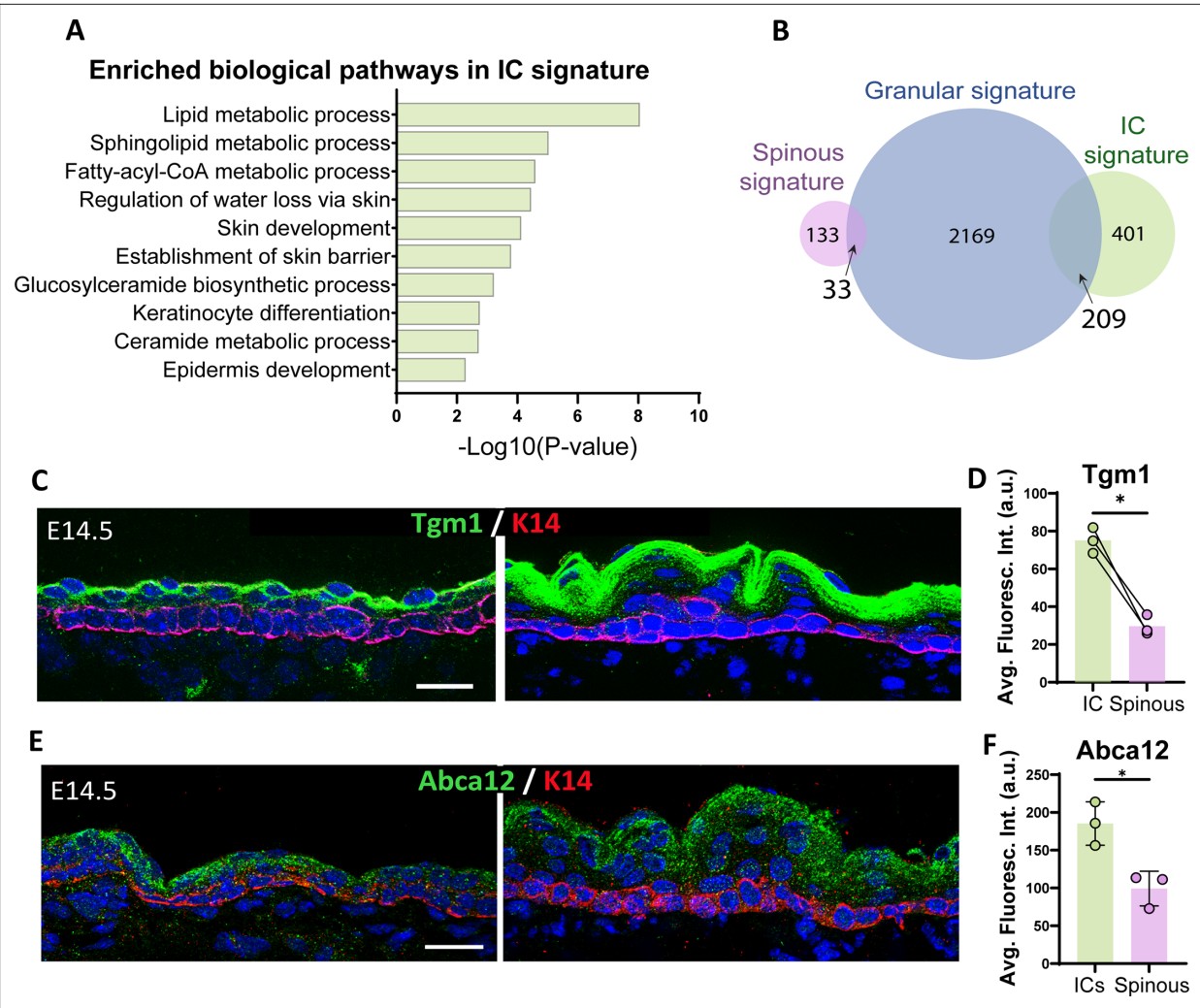

**Figure 4.** Intermediate cells (ICs) are transcriptionally similar to granular cells and express granular markers. (**A**) Gene Ontology (GO) term enrichment analysis of genes in the IC signature. (**B**) Venn diagram illustrating the number of genes in the IC signature compared with the granular signature (log2 fold change > 1, p<0.05) for each cell population. A total of 209 genes were commonly enriched in ICs and granular cells, which represents 34% of the total IC signature. (**C** and **E**) Immunofluorescence staining of Tgm1 (**C**) and Abca12 (**E**) in green and K14 (red) to mark basal cells in WT embryos at embryonic day (E) 14.5 vs E16.5. Scale bars: 20 µm. (**D**) Quantification of Tgm1 average fluorescence intensity at suprabasal cells at E14.5 (IC) vs the first two layers of suprabasal cells at E16.5 (spinous cells). n=3 embryos/time point. Bar represents the mean. *: p<0.05, two-tailed paired t-test. (**F**) Quantification of Abca12 average fluorescence intensity at suprabasal cells at E14.5 (ICs) vs the first two layers of suprabasal cells at E16.5 (spinous cells). n=3 embryos/time point. Data are presented as the mean ± SD. *: p<0.05, two-tailed unpaired t-test.

The online version of this article includes the following figure supplement(s) for figure 4:

**Figure supplement 1.** Intermediate and granular cells shared pathways related to lipid biosynthetic processes and endomembrane systems.

*2021*; GSE168011) revealed a striking overlap with over one-third of the genes in the IC signature also being part of the granular cell signature (*Figure 4B*). In contrast, the spinous gene signature overlap with granular cells was just under 20%. GO analysis of IC/granular shared genes revealed pathways such as keratinocyte differentiation, endomembrane system (reminiscent of lamellar bodies), and lipid processes (*Figure 4—figure supplement 1*). Notably, both *Grhl3* and *Hopx*, two transcription factors that control late epidermal differentiation, were upregulated in both of these cell types (*Chalmers et al., 2006*; *Chen et al., 2019*; *Obarzanek-Fojt et al., 2011*; *Ting et al., 2005a*; *Ting et al., 2005b*; *Yu et al., 2006*).

To validate the expression of canonical granular markers in ICs, we performed immunofluorescence staining of Tgm1, an enzyme that performs protein cross-linking of structural proteins for cornified envelope formation, and Abca12, which plays a role in transporting lipids into lamellar bodies and is

disrupted in harlequin ichthyosis (*Akiyama, 2010*; *Thomas et al., 2006*). Both proteins were already expressed at the protein level in ICs, but not in spinous cells (*Figure 4C–F*).

Together, these data suggested that in addition to promoting rapid amplification of keratinocytes, ICs may serve as progenitors for the first granular cells that form during development. To test this hypothesis, we performed a pulse-chase experiment in K10-rtTA;TRE-H2B-GFP embryos by injecting a low dose of doxycycline to induce the expression of H2B-GFP in K10-positive cells at E14.5 (i.e. only ICs) (*Figure 5—figure supplement 1A*). We followed the H2B-GFP+ cells over time, and we found that at E15.5, these cells were already more superficial, separated from basal cells by a layer of H2B-GFP-negative cells (*Figure 5A*). Based on their expression of MafB, these cells are the first spinous cells produced (*Figure 5A*). Strikingly, about 95% of H2B-GFP+ cells did not stain for MafB (*Figure 5A'*). Thus, rather than converting to a spinous cell fate, they are displaced upward. When we chased these cells until E16.5, they were located in the uppermost layers of the epidermis (*Figure 5B*), and they co-stained with loricrin, a granular cell marker (*Figure 5—figure supplement 1A*). This indicates that most ICs transition into granular cells without going through a MafB+ state. When labeled cells were chased until E18.5, GFP signal was clearly present in the cornified layer, demonstrating that the traced ICs had terminally differentiated before birth (*Figure 5C*). The above data suggest that the first spinous cells arise from basal cells and not from differentiation of ICs. To test this, we generated K14-rtTA;TRE-H2B-GFP mice and pulsed them with low levels of doxycycline at E14.5. With these conditions, we specifically labeled basal cells at E14.5 and followed their progeny. In embryos chased to E15.5, we observed that 80% of basal cells and 70% of all MafB+ cells were H2B-GFP+, demonstrating that most spinous cells arise from basal divisions (*Figure 5D and D'*). Altogether, these data demonstrate that most ICs are direct precursors for granular cells and do not pass through a MafB+ spinous intermediate (*Figure 5—figure supplement 2*).

## Contractility status changes through epidermal development and is sufficient to drive aspects of granular cell fate

Further analysis of our RNA-Seq dataset suggested that ICs may be mechanically, as well as molecularly, distinct from spinous cells. We examined the list of genes that were upregulated in ICs vs spinous cells, regardless of their expression levels in basal cells (*Figure 6A*). GO term analysis revealed that regulation of cell morphogenesis and the actin cytoskeleton were among the most highly upregulated pathways in ICs (*Figure 6A*). Heatmap analysis of the contractome gene set *Zaidel-Bar et al., 2015* demonstrated clear differences in suprabasal cell expression at E14.5 and E16.5 (*Figure 6B*). To validate these findings, we stained for markers that indicate contractility status. We observed that F-actin levels were higher in basal and suprabasal cells at E14.5, as well as in granular cells at E16.5. However, they were low in spinous cells at this later time point (*Figure 6C–C''*). Myosin IIA, a major contributor to actomyosin contractility in the epidermis (*Miroshnikova et al., 2018*; *Sumigray et al., 2012*), was also higher in ICs than spinous cells (*Figure 6D–D''*). Similarly, α18, an antibody that recognizes an epitope of α-catenin that is exposed when adherens junctions are under tension (*Yonemura et al., 2010*), was also higher in basal and suprabasal cells at E14.5 vs E16.5 (*Figure 6E–E''*). Furthermore, active nuclear YAP, a mechanoresponsive transcriptional co-activator, was present in ICs and granular cells at E16.5, but not in spinous cells (*Figure 6—figure supplement 1*). Similarly, the percentage of basal cells at E14.5 with active YAP was significantly higher than basal cells at E16.5. Apart from its role in mechanotransduction, the active form of YAP is an important driver of proliferation in the basal layers of the epidermis (*Aragona et al., 2020*; *Schlegelmilch et al., 2011*; *Zhang et al., 2011a*). To test a potential role of YAP in IC proliferation, we used K10-rtTA;TRE-YAP1$^{S112A}$-GFP (hereafter called K10-YAP$^{CA}$) mice to induce an active form of YAP, marked by H2B-GFP, in suprabasal cells at E16.5. Active YAP was not sufficient to induce proliferation in spinous cells; however, it resulted in a non-cell autonomous increase in proliferation of basal cells (*Figure 6—figure supplement 2*).

The apparent changes in contractility through epidermal development raised the question of whether the shared contractility between ICs and granular cells lies downstream of differentiation, or if it contributes to their similar gene expression. To test the effects of heightened contractility in the suprabasal cells, we turned to mouse models we previously developed that allow doxycycline-induced actomyosin contractility: K10-rtTA;TRE-Spastin and K10-rtTA;TRE-Arhgef11$^{CA}$, hereafter referred to as K10-Spastin and K10-Arhgef11$^{CA}$, respectively (*Hinnant et al., 2024*; *Muroyama and Lechler, 2017*; *Muroyama et al., 2018*; *Ning et al., 2021*). In both cases, this expression is confined

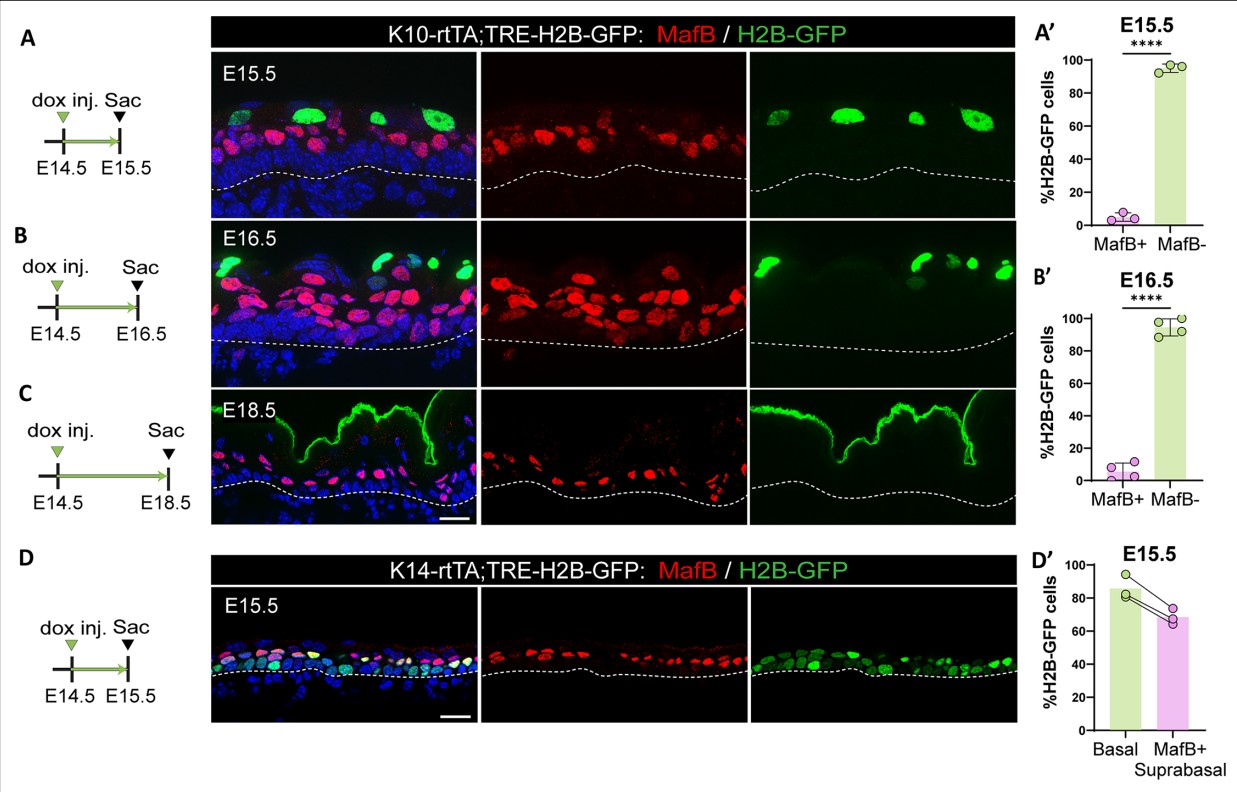

**Figure 5.** Intermediate cells (ICs) do not differentiate into spinous cells. (**A**, **B**, and **C**) Immunofluorescence staining of spinous cells with MafB in red and labeled cells with H2B-GFP from K10-rtTA;TRE-H2B-GFP mice that were injected with a low dose of doxycycline at embryonic day (E) 14.5 to label only ICs and sacrificed at E15.5 (**A**), E16.5 (**B**), or E18.5 (**C**). Basement membrane is indicated as a dotted line. Scale bars: 20 µm. (**A'** and **B'**) Percentage of MafB+ and MafB- suprabasal cells expressing H2B-GFP at E15.5 (n=3 embryos) (**A**) and E16.5 (n=4 embryos) (**B**). Data are presented as the mean ± SD. ****: p<0.0001, two-tailed unpaired t-test. (**D**) Immunofluorescence staining of spinous cells with MafB in red and labeled cells with H2B-GFP in K14-rtTA;TRE-H2B-GFP mice that were injected with a low dose of doxycycline at E14.5 and sacrificed at E15.5. Scale bar: 20 µm. (**D'**) Percentage of basal cells and MafB+ suprabasal cells expressing H2B-GFP. Paired samples are quantifications from the same mouse (n=3 K14-rtTA;TRE-H2B-GFP embryos).

The online version of this article includes the following figure supplement(s) for figure 5:

**Figure supplement 1.** Intermediate cells differentiate into granular cells.

**Figure supplement 2.** Updated model of early epidermal stratification.

to suprabasal cells. K10-Spastin expresses the active form of the microtubule severing protein Spastin which increases contractility (***Ning et al., 2021***); and K10-Arhgef11$^{CA}$ induces expression of a constitutively active Rho-GEF that activates RhoA GTPase, a key positive regulator of actomyosin contractility (***Ning et al., 2021***).

We started by examining the transcriptomic changes that are induced by contractility (from K10-Spastin vs controls at E16.5, log2 fold change ≥ 1) and compared it with the IC and granular cell signatures. Remarkably, we found that 47% of contractility-induced genes overlapped with the granular gene signature (***Figure 7—figure supplement 1A***), demonstrating a major change in gene expression and differentiation state by intracellular contractility (hypergeometric test representation factor 3.4, p=2 × 10$^{-101}$). GO analysis revealed that the common signature was highly enriched in keratinocyte differentiation, cornified envelope, lamellar bodies, and lipid processes – all hallmarks of granular cells (***Figure 7A***). When comparing with the IC signature, there was still significant overlap – 80 of the 610 genes in the IC signature (13%) were also found in the contractile signature, though less than that found in the granular cells (***Figure 7—figure supplement 1B***). These included the late epidermal differentiation transcriptional regulators *Grhl3* and *Hopx*. The most enriched terms for the overlapping genes between ICs and the contractile gene signature were lipid processing and transport (***Figure 7B***). In addition, mainly lipid metabolic genes were upregulated in the shared ICs, granular cells, and contractility signatures (***Figure 7—figure supplement 1C***), suggesting that these

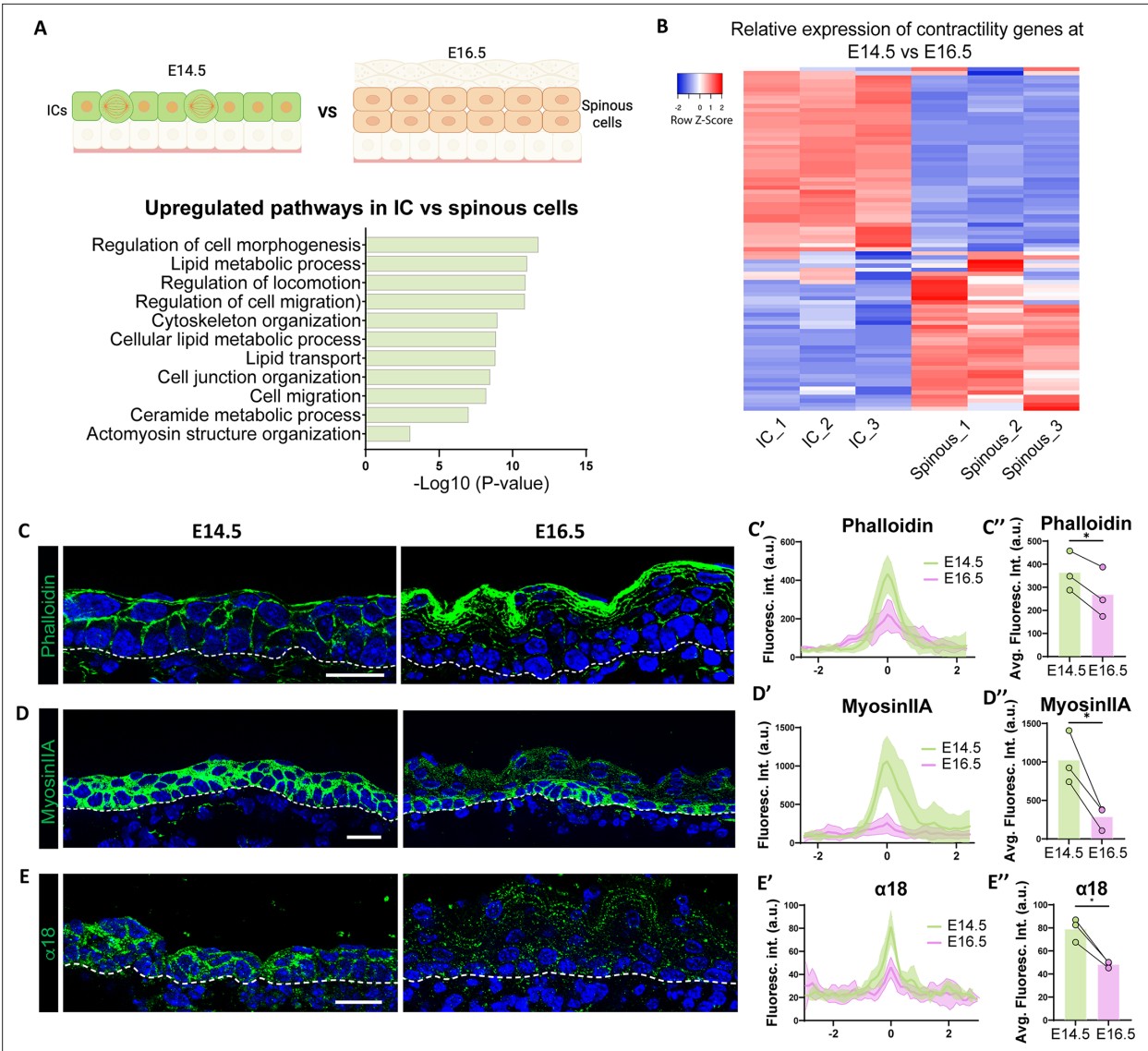

**Figure 6.** Intermediate cells show upregulation of contractility markers as compared to spinous cells. (**A**) Gene Ontology (GO) term enrichment analysis of genes upregulated in intermediate cells vs spinous cells at embryonic day (E) 16.5, as shown in the above diagram, regardless of their expression in basal cells (log2 fold change > 1, p<0.05) reveals upregulation of biological processes related to cytoskeleton organization. Created with BioRender. com. (**B**) RNA-Seq heatmap depicting differential expression of contractome genes in intermediate and spinous cells at E16.5. Gene expression by FPKM was log2-transformed. (**C**, **D**, and **E**) Immunofluorescence staining of Phalloidin (**C**), Myosin IIA (**D**), and α18 (**E**) in green at E14.5 vs E16.5. Basement membrane is indicated as a dotted line. Scale bars: 20 μm. (**C'**, **D'**, and **E'**) Fluorescence intensity of suprabasal-suprabasal cell boundaries in the first two layers of suprabasal cells. Line scans were performed in WT embryos at E14.5 (between intermediate cells) and E16.5 (between spinous cells). Line scan graphs show centered measurements across six cell-cell boundaries at each time point. Data are presented as the mean ± SD (shown as colored shadows in green at E14.5 and pink at E16.5). Scale bars: 20 μm. (**C''**, **D''**, and **E''**) Quantification of average fluorescence intensity from the line scan maximum values at suprabasal-suprabasal cell boundaries in WT E14.5 and E16.5. For bar plots, bars represent the mean, joined dots are paired samples from each time point. n=3 embryos/time point, average maximum values of at least 15 line scans at suprabasal cell boundaries in each embryo were calculated. *: p<0.05, two-tailed paired t-test.

The online version of this article includes the following figure supplement(s) for figure 6:

**Figure supplement 1.** Like granular cells, more intermediate cells have nuclear YAP than spinous cells.

**Figure supplement 2.** Inducing constitutively active YAP in spinous cells does not increase their proliferation.

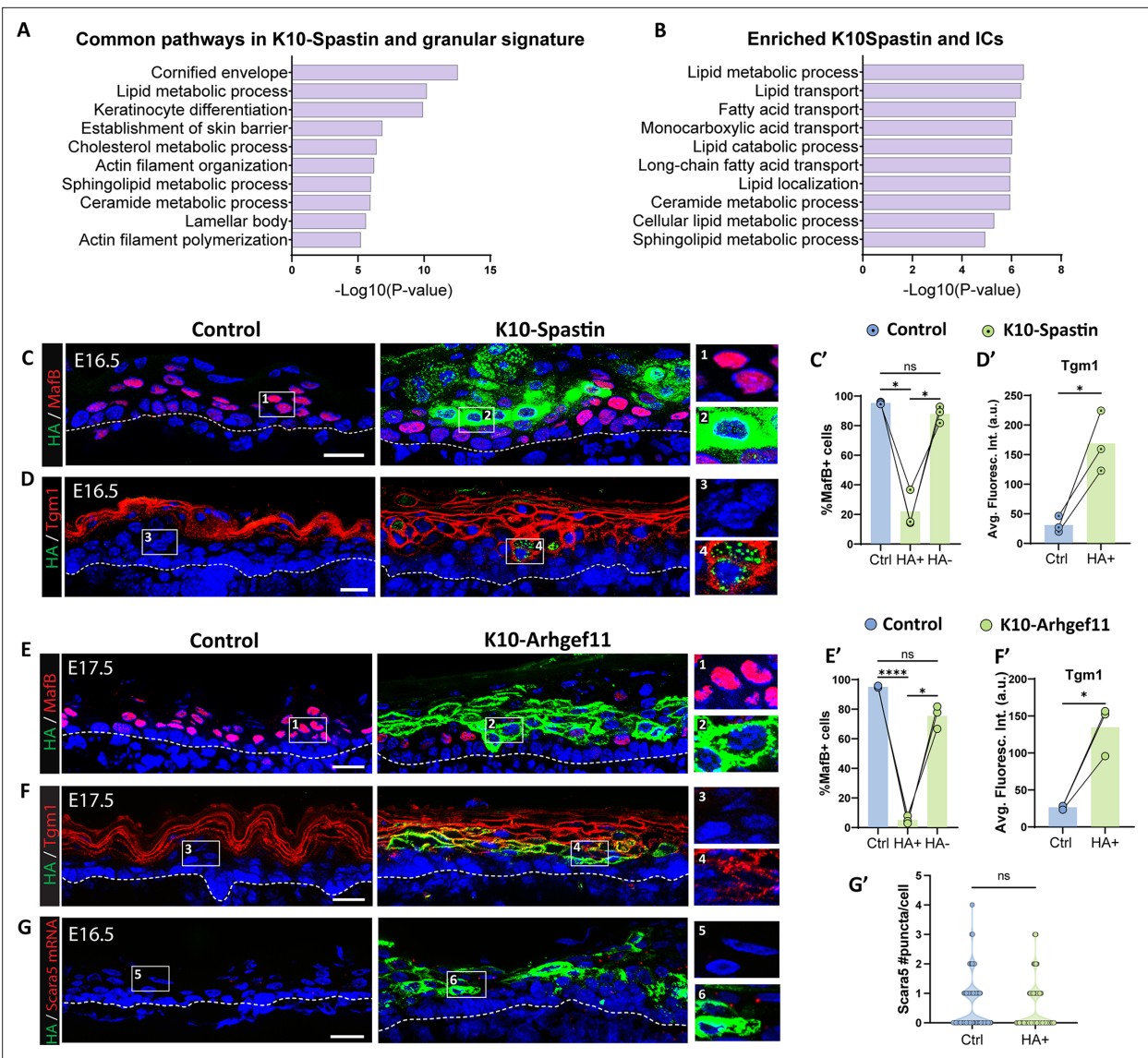

**Figure 7.** Increased contractility in spinous cells is sufficient to induce many granular genes. (**A** and **B**) Gene Ontology (GO) term analysis of the list of genes commonly upregulated in suprabasal K10-Spastin (vs control spinous cells at embryonic day [E] 16.5, log2 fold change > 1, p<0.05) and granular cell signature (**A**) or intermediate cell signature (**B**). (**C** and **D**) Immunofluorescence staining of MafB (**C**) or Tgm1 (**D**) in red, and suprabasal cells expressing Spastin marked by HA (green) in K10-Spastin and controls. Images of E16.5 embryos, doxycycline fed starting at E14.5. Insets to the left of all images in this panel show close-ups of HA+ cells in suprabasal layers adjacent to basal cells. Scale bars: 20 µm. (**E** and **F**) Immunofluorescence staining of MafB (**F**) or Tgm1 (**H**) in red, and suprabasal cells expressing Arhgef11$^{CA}$ marked by HA (green) in K10-Arhgef11 and controls. Images of E17.5 embryos, doxycycline fed from E10.5. Scale bars: 20 µm. (**C'** and **E'**) Percentage of MafB+ suprabasal cells in controls, and in HA+ and HA- cells in K10-Spastin at E16.5 (**C'**) or K10-Arhgef11$^{CA}$ at E17.5 (**E'**). Total number of mice analyzed was three embryos per genotype. Data are presented as the mean ± SD. *: p<0.05, ****: p<0.0001, repeated-measures one-way ANOVA (p=0.0174 (**C'**) and p=0.0041 (**E'**)), Tukey's multiple comparisons test. For all plots in this panel, cells located in the first two layers of suprabasal cells immediately above the basal layer were considered for quantification. (**D'** and **F'**) Quantification of average fluorescence intensity of Tgm1 in suprabasal cells of control and HA+ cells in K10-Spastin at E16.5 (**D'**) or K10-Arhgef at E17.5 (**F'**). Total number of mice analyzed was three embryos per genotype. *: p<0.05, two-tailed paired t-test. (**G**) RNAscope of *Scara5* in red and immunostaining with HA in green in control and K10-Arhgef11. Images of E16.5 embryos, doxycycline fed starting at E14.5. Scale bars: 20 µm. (**G'**) Violin plot of number of Scara5 puncta per suprabasal cell in control, and HA+ cells and neighbor HA- cells in K10-Spastin at E16.5. n=224 HA+ cells and 160 WT cells from two embryos per genotype. ns: not significant, Mann-Whitney test.

The online version of this article includes the following figure supplement(s) for figure 7:

**Figure supplement 1.** K10-Spastin cells are more granular-like than intermediate-like, and lipid metabolic processes are commonly upregulated among K10-Spastin, intermediate cells (ICs), and granular cells.

**Figure supplement 2.** Increased contractility in spinous cells of adult epidermis induces some granular markers in adult epidermis.

*Figure 7 continued on next page*

*Figure 7 continued*

**Figure supplement 3.** Increased contractility in spinous cells accelerates acquisition of granular markers.

**Figure supplement 4.** Inducing constitutively active YAP in spinous cells does not induce granular markers.

**Figure supplement 5.** Increased contractility does not alter H3K27me3 levels.

were the pathways commonly upregulated in ICs and granular cells as a consequence of contractility. Notably, many EDC genes were enriched in the contractile and granular signatures, but many fewer in the IC signature (*Supplementary file 2*). This demonstrates specific differentiation pathways are enriched in ICs, while the contractile gene signature is much more similar to the full granular gene expression program.

In contrast to an upregulation of granular gene markers in hypercontractile skin, we noted that about one-third of all the spinous signature genes were downregulated in this mutant (55/163 genes), demonstrating a partial suppression of the spinous cell fate.

We next examined the effects of increased contractility through analysis of specific markers for spinous cells we identified earlier in this study. We used both models of increased contractility in supra-basal cells: K10-Spastin and K10-Arhgef11CA, where expression of the transgene for both is detectable by HA staining. In both mouse models, suprabasal cells with increased contractility (at E16.5 and later) did not express the spinous marker MafB (*Figure 7C and E*), even when they were positioned in the first layers on top of basal cells, where they should clearly be spinous cells. In contrast, these cells precociously expressed the granular marker Tgm1 (*Figure 7D and F*). Similar results were obtained in back and glabrous skin of K10-Arhgef11CA adult mice (*Figure 7—figure supplement 2*), and at E17.5, when suprabasal contractility was induced after initial spinous cell specification (*Figure 7—figure supplement 3*). These latter results suggest that the effects of increased suprabasal contractility are the same regardless of the time of its induction.

Analysis of the IC marker *Scara5* revealed that increasing suprabasal contractility did not alter *Scara5* mRNA levels in HA+ cells compared to controls, suggesting that contractility induces a granular-like rather than intermediate-like cell type in mutant mice (*Figure 7G*).

YAP concentrates within the nucleus in suprabasal epidermal cells in both of our mouse models that increase contractility (*Ning et al., 2021*), and YAP is also localized to the nucleus in granular cells (*Figure 6—figure supplement 1*). To test the role of YAP in inducing a granular-like state, we again used the mouse model K10-YAPCA. We observed that inducing active YAP did not induce Tgm1 in spinous cells and did not suppress the expression of MafB, the spinous cell marker (*Figure 7—figure supplement 4*). These data suggest that YAP activation lies downstream of contractility but is not sufficient to induce the effects observed by increased contractility in spinous cells. In addition to YAP activation, stretching of keratinocytes results in epigenetic changes, including an increase in H3K27me3 levels (*Le et al., 2016*). However, inducing contractility did not cause a significant change in nuclear levels of this epigenetic mark (*Figure 7—figure supplement 5*).

## Discussion

Embryonic epidermal cells coordinate their growth and differentiation to allow rapid stratification and barrier formation. Our data here characterize ICs in epidermal development, revealing novel trajectories of cell differentiation. Further, our data directly demonstrate that actomyosin contractility in the developing epidermis can play an instructive role in differentiation, promoting granular-like gene expression and suppressing spinous markers.

Bulk RNA-Seq analysis has allowed us to characterize embryonic epidermal cells in greater depth. This revealed that basal cells undergo significant changes between E14.5 and E16.5, consistent with the fact that they generate distinct cell progeny at these time points (ICs vs spinous cells). In addition, like suprabasal cells, basal cells at these time points also show differences in mechanical properties. Whether this is in response to changes in their physical environment or is mediated by internal timers remains unknown. That said, understanding specific markers and transcriptional regulators of these maturing basal cells may allow us to manipulate these transitions to maintain or bypass distinct stages of epidermal development.

Analysis of the IC transcriptome revealed an unexpected expression of granular genes in these transient cells. This gene expression program was especially enriched for lipid-modifying enzymes

that play a key role in barrier formation. Lipids are synthesized in granular cells and stored in lamellar bodies before secreting them to the intercellular spaces (*Prado-Mantilla and Lechler, 2023*; *Vietri Rudan and Watt, 2021*). Our data suggest that this process may begin early in ICs. In contrast, few EDC genes were upregulated in ICs, suggesting that only parts of the granular program are expressed early in these cells, and many of the structural components of the eventual barrier are expressed only later.

Further, our short-term lineage tracing suggests that most ICs transition into granular cells without going through a MafB+ (a spinous cell marker) stage. This reveals two distinct differentiation pathways leading to the granular cell state, one through ICs during development and one through spinous cells postnatally. This raises questions about whether each route will require distinct regulatory and transcriptional pathways.

Finally, our data revealed mechanical similarities between ICs and granular cells, in addition to their similarities at the transcriptomic level. Actomyosin regulators were transcriptionally upregulated in both of these cell types as compared to spinous cells, and we were able to validate these findings in intact skin. Importantly, we found that contractility was sufficient to induce granular-like gene expression and to repress some spinous cell markers. This reveals a rather direct effect of contractility lying upstream, as well as presumably downstream, of differentiation pathways. It also explains the observation that increased contractility results in premature formation of the epidermal barrier (*Muroyama and Lechler, 2017*; *Ning et al., 2021*). It has been appreciated for some time that granular cells have increased actomyosin contractility, which is important for the function and placement of tight junctions specifically in this layer of the epidermis (*Itoh et al., 2012*; *Prado-Mantilla and Lechler, 2023*; *Rübsam et al., 2017*; *Sumigray et al., 2012*). There are many intriguing possibilities for how contractility may affect cell fates, such as mechanosensitive transcriptional regulators that mediate this effect. We tested a possible role for YAP1, a transcriptional regulator whose activation is induced by contractility in the epidermis. We found that active YAP1 was insufficient to drive these gene expression changes. Future work is required to determine the mechanism of mechanosensitive differentiation in the epidermis. The epidermis increases contractility in response to externally applied forces that result in stretch and in response to perturbation of cell adhesions (*Aragona et al., 2020*; *Sumigray et al., 2014*). It will be important to test whether these stimuli alter differentiation pathways through their effects on contractility.

Of note, we found that contractility had a strong effect on the EDC, whose positioning within the nucleus changes with differentiation (*Gdula et al., 2013*; *Mardaryev et al., 2014*; *Williams et al., 2002*). This suggests that aspects of epigenetics and chromatin organization may be downstream of contractility, as shown by external forces directly stretching chromatin (*Tajik et al., 2016*) and extrinsic topographical cues constricting the nucleus to induce chromatin reprogramming (*Wang et al., 2023*). Notably, we found that there is a decrease in mRNA levels of Sun1 and Nesprin1/3, parts of a complex that tether the actin cytoskeleton in the cytoplasm to chromatin in the nucleus when contractility is increased, in hypercontractile cells. Loss of Sun1/2 also leads to precocious differentiation in the epidermis (*Carley et al., 2021*) through effects on integrin-mediated mechanical integration, though whether regulation of these connections mediates the contractility effect seen in suprabasal cells will require further investigation.

## Materials and methods
### Mice
All animal work was approved by Duke University's Institutional Animal Care and Use Committee and performed in accordance with their committee guidelines. Mice were genotyped by PCR, and both male and female mice were analyzed in this study. All mice were maintained in a barrier facility with 12 hr light/dark cycles. Mouse strains used in this study include K14-RFP (*Zhang et al., 2011b*), K10-rtTA and TRE-Spastin (*Muroyama and Lechler, 2017*), K14rtTA (*Nguyen et al., 2006*) and TRE- Arhgef11$^{CA}$ (*Ning et al., 2021*), CD1 (Charles River, strain code: 022). Other mice were from the Jackson Laboratories, and their stock numbers are as follows: TRE-H2B-GFP (005104), TRE-YAP1$^{S112A}$-GFP (031279; *Gao et al., 2013*).

### Generation of the TRE-MafB-HA mouse line
MafB with an HA tag on the C-terminus followed by a stop codon was synthesized by GenScript and ligated into the pTRE2 vector. To verify the proper doxycycline-dependent expression of the

TRE-MafB-HA cassette, the vector was co-transfected with a K14-rtTA plasmid into cultured keratinocytes and placed in doxycycline-containing media. TRE-MafB-HA plasmid was linearized, purified, and used by the Duke Transgenic Core to generate transgenic TRE-MafB-HA mice via pronuclear injection.

## Skin explant preparation

Back skin of embryonic mouse skin was dissected, rinsed, and gently unfolded in sterile PBS, then placed on a 2% agarose pad diluted in a 1:1 sterile water:media mixture (10% FBS in DMEM, doxycycline [2 µg/ml, to induce H2B-GFP expression] and 1:100 penicillin/streptomycin). Note that the dermis of the skin explant faces the agarose pad. The skin explants on agarose were gently placed into a 6 cm dish containing 1 ml 10% FBS in DMEM (Gibco, 11965). The explants were cultured at 37°C and 7.5% $CO_2$ for at least 2 hr for recovery before live imaging.

## Sample preparation for RNA-Seq

K10-rtTA;TRE-H2B-GFP;K14-RFP pregnant dams were all fed with doxycycline chow at E12.5, and then sacrificed at either E14.5 or E16.5. Embryos were checked under a dissecting microscope for GFP and RFP expression before use, and their tails were taken to confirm genotypes. E16.5 and E14.5 back skins were cut off and treated with 1.4 U/ml dispase II (Roche, 4942078001) in HBSS at room temperature for 1 hr, then peeled off the epidermis from the dermis. The epidermis of E14.5 back skin was not peeled off due to its low thickness, but it was still treated with dispase II to maintain the same conditions as the E16.5 samples. Both E16.5 epidermis and E14.5 skin samples were digested in 1:1 trypsin (GIBCO, 25200-056) with versene (Gibco, 15040-066) at 37°C and rotated for 20 min. Samples were then mixed 1:1 with FACS buffer (HBSS with 2.5% FBS and 10 µg/ml DNAase I) and centrifuged at low speed. The cell pellet was diluted into FACS buffer with propidium iodide solution (Sigma, P4864, to exclude dead cells) and filtered using sterile CellTrics 30 µm filters (Sysmex, 04-004-2326). The lectin UEA1 (Ulex europaeus agglutinin I) binds to carbohydrates in cell surfaces present in uppermost keratinocytes. To exclude them, we used biotinylated UEA1 (Vector Laboratories, B-1065) and detected them with APC-streptavidin (BioLegend, 405207). The E14.5 ICs (K10-GFP$^+$;K14-RFP$^-$;APC-UEA1$^-$), E16.5 spinous cells (K10-GFP$^+$;K14-RFP$^-$;APC-UEA1$^-$), and basal cells at E14.5 and E16.5 (K10-GFP$^-$;K14-RFP$^+$) cells were FACS-sorted separately. RNA was extracted using a QIAGEN RNeasy Mini Kit (QIAGEN, 74104) following the manufacturer's protocols. Genomic DNA was removed using RNase-Free DNase (QIAGEN, 79254). Biological replicates, three independent RNA samples from each cell population, were collected and sent for sequencing and analysis by Novogene.

## RNA-Seq analysis

Reads were trimmed and aligned to the mouse reference genome mm10 (GRCm38) using STAR (v2.5). Gene-level quantification was performed using HTSeq (v0.6.1), and expression values were calculated as FPKM. Differential gene expression analysis between two time points or cell types (three biological replicates per group) was performed using the DESeq2 R package (v1.6.3), with p-values adjusted for false discovery using the Benjamini-Hochberg method. The gene signature of a cell population was obtained by the overlapping genes that were enriched compared to the suprabasal cells at a different time point and enriched compared to the basal cells at its corresponding time point. A significance threshold of adjusted p-value<0.05, log2 fold change ≥ 1, and FPKM ≥ 1 was used for defining marker genes of each cell population.

GO term analysis for differentially expressed genes and gene signatures was performed using the GO enrichment analysis (*Ashburner et al., 2000*; *Aleksander et al., 2023*; *Thomas et al., 2022*).

A volcano plot was generated using the ggplot2 package in R. Genes were classified as upregulated in each cell population based on thresholds of |log2 fold change| > 1 and adjusted p-value<0.05. Threshold lines were added at these cutoffs. Candidate marker genes were highlighted and labeled using ggrepel.

FPKM of differentially expressed genes in the contractome list was log2-transformed and used for heatmaps. Heatmaps were generated using the online resource Heatmapper (*Babicki et al., 2016*), using the average linkage clustering method and Pearson's distance measurement method. Raw data can be found in GEO – GSE295753.

PCA plot of all cell population data was generated using the web tool Clustvis: https://biit.cs.ut.ee/clustvis/ (*Metsalu and Vilo, 2015*).

Granular cell signature was determined from genes significantly upregulated in ETA sheets vs epidermal sheets (p-value<0.05, log2 fold change ≥ 1) obtained from the previously published dataset: GSE168011 (*Matsui et al., 2021*).

## Immunofluorescence

Fresh tissue was embedded in OCT (Sakura), frozen, and sectioned at 10 μm using a cryostat. Sections were fixed with 4% paraformaldehyde (PFA) in PBS for 8 min at room temperature or ice-cold acetone (for Tgm1 and St8sia6 staining) for 2 min, washed with PBS containing 0.2% Triton (PBST) for 5 min, then blocked with blocking buffer (3% bovine serum albumin with 5% normal goat serum [Gibco, 16210064], and 5% normal donkey serum [Sigma-Aldrich, D9663] in PBST) for 15 min. Sections were incubated with primary antibodies diluted in blocking buffer for 1 hr at room temperature (α18 antibody was incubated for 15 min), followed by three washes with PBST, and incubated in secondary antibodies and stains, such as Hoechst 34580 or Phalloidin, for 15 min at room temperature. After three washes with PBST, sections were finally mounted in the anti-fade buffer (90% glycerol in PBS plus 2.5 mg/ml *p*-phenylenediamine [Thermo Fisher, 417481000]) and sealed using clear nail polish along the borders.

Primary antibodies used in this study: rat anti-HA (Sigma-Aldrich, 11867423001), chicken anti-keratin 5/14 (generated in the Lechler lab), rabbit anti-K10 (Covance, 905401), guinea pig anti-K10 (Progen, GP-K10), rat anti-β4 integrin (BD Biosciences, 553745), rabbit anti-Myosin IIA (BioLegend, PRB-440P), rat anti-α18 (gift from Akira Nagafuchi, Kumamoto University), rabbit anti-YAP/TAZ (Cell Signaling Technology, 8418S), rabbit anti-Tgm1 (Proteintech, 12912-3-AP), rabbit anti-MafB (Novus, NBP1-81342), rabbit anti-St8sia6 (Sigma, HPA011635), rabbit anti-Abca12 (gift from Dr. Wong, University of Michigan). F-actin was stained with Phalloidin-488 (Invitrogen, A12379).

## EdU labeling

Pregnant dams were intraperitoneally injected with 10 mg/kg of EdU and sacrificed after 1 hr for tissue dissection. Back skins of embryos were collected, and tails were taken for genotyping. Tissue sections were fixed with 4% PFA and stained with primary and secondary antibodies, then EdU was detected following Click-iT EdU (Thermo Fisher, C10337) protocol.

## Lineage tracing/pulse chase

For IC lineage tracing: K10-rtTA;TRE-H2B-GFP pregnant dams were intraperitoneally injected with a low dose of doxycycline (0.5 mg/kg) at E14.5 to label only ICs. Dams were sacrificed, and their embryos were collected after 1 day (E15.5), 2 days (E16.5), or 4 days (E18.5) for tissue dissection. Their tails were kept for genotyping and K10-rtTA;TRE-H2B-GFP embryos were sectioned and analyzed.

For basal cell lineage tracing: K14-rtTA;TRE-H2B-GFP pregnant dams were intraperitoneally injected with doxycycline (50 mg/kg) at E14.5 to label basal cells. Dams were sacrificed, and their embryos were collected after 1 day, at E15.5.

## RNAscope

RNAscope was performed using the Multiplex Fluorescent v1 and v2 systems (ACD, 323100) followed by antibody co-staining. Back skins from mouse embryos were freshly frozen in OCT and sectioned at 10 μm. Tissue sections were fixed for 1 hr with 4% PFA at 4°C. After fixation, standard RNAscope protocols were used according to the manufacturer's instructions. The following probes were used: St8sia6 (ACD, 887831-C1) and Scara5 (ACD, 522301-C1). TSA Vivid Fluorophore 570 (ACD, 323272) was used to develop probe signal. Then, antibody staining was performed to quantify the probe signal in suprabasal cells, using guinea pig anti-K10 antibody, and in basal cells, using chicken anti-K5/14. HA+ cells were marked with rat anti-HA or rabbit anti-MafB (for K10-MafB samples). Coverslips were mounted using Prolong Gold (Invitrogen, P10144).

## Imaging

For live imaging of K10+ cell division, prepared skin explants were placed upside down in a Lumox dish 35 (Sarstedt, 94.6077.331) with the epidermal side facing toward the membrane. Samples were imaged at 15 min intervals overnight using the MetaMorph software on an Andor XD revolution spinning disc confocal microscope at 37°C and 5% $CO_2$ using a ×20/0.5 UplanFl N dry objective.

For immunofluorescence staining, tissue sections were imaged on a Zeiss AxioImager Z1 microscope with Apotome.2 attachment, Plan-APOCHROMAT ×20/0.8 objective, Plan-NEOFLUAR ×40/1.3 oil objective, or Plan-NEOFLUAR ×63/1.4 oil objective, Axiocam 506 mono camera, and acquired using Zen software (Zeiss). When making intensity measurement comparisons, all images within one experiment were taken with identical exposure times.

## Image quantification and statistics

All image quantifications were done using Fiji software. Quantifications of fluorescence intensity of cortical F-actin, Myosin IIA, and α18 were measured by drawing 10-pixel wide lines across cell-cell boundaries. Maximum values from plot profiles were aligned to yield the final line scan plots. The mean of the maximum values of each line scan was calculated per mouse and compared between conditions or time points to determine statistical significance.

EdU, Scara5, St8sia6, YAP, and MafB measurements were determined by calculating the percentage of the number of positive cells in the total number of suprabasal cells in the first two suprabasal layers adjacent to basal cells. Quantifications are from at least three fields per mouse.

Tgm1 average fluorescence intensity was calculated by measuring the mean value of suprabasal cells above basal cells. This area was defined by drawing 50-pixel wide lines in the suprabasal area, along the labeled basal layer, using the Freehand Line tool in Fiji.

All statistical analyses were performed using GraphPad Prism 10 software.

Data shown in bar plots was presented as mean ± standard deviation (SD), and significance was determined using two-tailed Student's t-test, or, for multiple comparisons, one-way or two-way ANOVA, followed by Tukey's or Sidak's tests. Further details in statistical analysis were specified in figure legends. Data were determined to be statistically significant when p-value<0.05. Asterisks denote statistical significance (ns = not significant, *: $p<0.05$, **: $p<0.01$, ***: $p<0.001$, ****: $p<0.0001$).

## Figure graphics

Figure graphics were created in BioRender (https://BioRender.com/uykhkg9).

## Acknowledgements

We thank Julie Underwood for expert care of our mice and lab management as well as members of the Lechler Lab for comments on the manuscript. Maggie Bara provided initial characterization of the TRE-MafB line, David Kirsch provided mouse lines, and Sunny Wong provided Abca12 antibody.

## Additional information

### Funding

| Funder | Grant reference number | Author |
| --- | --- | --- |
| NIH-NIAMS | R01AR08108 | Terry Lechler |
| NIH-NIAMS | R01AR067203 | Terry Lechler |
| NIH-NIAMS | R01AR083352 | Terry Lechler |

The funders had no role in study design, data collection and interpretation, or the decision to submit the work for publication.

### Author contributions

Alexandra Prado-Mantilla, Wenxiu Ning, Data curation, Formal analysis, Investigation, Writing – original draft, Writing – review and editing; Terry Lechler, Conceptualization, Supervision, Writing – original draft, Project administration, Writing – review and editing

### Author ORCIDs

Wenxiu Ning ⓘ https://orcid.org/0000-0003-1028-8151
Terry Lechler ⓘ https://orcid.org/0000-0003-3901-7013

### Ethics

This study was performed in accordance with the Guide for the Care and Use of Laboratory Animals of the National Institutes of Health and with Duke IACUC protocol A255-23-12.

Reviewer #1 (Public review): https://doi.org/10.7554/eLife.100961.3.sa1
Reviewer #2 (Public review): https://doi.org/10.7554/eLife.100961.3.sa2
Reviewer #3 (Public review): https://doi.org/10.7554/eLife.100961.3.sa3
Author response https://doi.org/10.7554/eLife.100961.3.sa4

## Additional files

### Supplementary files

Supplementary file 1. Table of gene signatures.

Supplementary file 2. Epidermal differentiation complex gene expression levels.

### Data availability

Raw data files for the RNA -sequencing analysis have been deposited in the NCBI Gene Expression Omnibus under accession number GEO: GSE295753.

The following dataset was generated:

| Author(s) | Year | Dataset title | Dataset URL | Database and Identifier |
| --- | --- | --- | --- | --- |
| Prado-Mantilla A, Ning W, Lechler T | 2025 | Molecular and Mechanical Signatures Contributing to Epidermal Differentiation and Barrier Formation | https://www.ncbi.nlm.nih.gov/geo/query/acc.cgi?acc=GSE295753 | NCBI Gene Expression Omnibus, GSE295753 |

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
