## [Editor Report · eLife Assessment]

The authors address a **fundamental** question for cell and tissue biology. They use the skin epidermis as a paradigm and ask how stratifying self-renewing epithelia induce differentiation and upward migration in basal dividing progenitor cells to generate suprabasal barrier-forming cells that are essential for a functional barrier formed by such an epithelium. The authors provide **compelling** evidence time that an increase in intracellular actomyosin contractility, a hallmark of barrier-forming keratinocytes, is sufficient to trigger terminal differentiation, providing in vivo evidence of the interdependency of cell mechanics and differentiation. To illustrate their points, the authors use a combination of genetic mouse models, RNA sequencing, and immunofluorescence analysis. Precisely how the changes in gene expression, cell morphology, mechanics, and cell position are instructive and whether consecutive changes in differentiation are required still remain unclear, but the paper takes a nice step in advancing our knowledge of the process.

---

## [Referee Report · Reviewer #1 (Public review)]

Summary:

The authors address a fundamental question for cell and tissue biology using the skin epidermis as a paradigm and ask how stratifying self-renewing epithelia induce differentiation and upwards migration in basal dividing progenitor cells to generate suprabasal barrier-forming cells that are essential for a functional barrier formed by such an epithelium. The authors show for the first time that an increase in intracellular actomyosin contractility, a hallmark of barrier-forming keratinocytes, is sufficient to trigger terminal differentiation. Hence the data provide in vivo evidence of the more general interdependency of cell mechanics and differentiation. The data appear to be of high quality and the evidences are strengthened through a combination of different genetic mouse models, RNA sequencing and immunofluorescence analysis.

To generate and maintain the multilayered, barrier-forming epidermis, keratinocytes of the basal stem cell layer differentiate and move suprabasally accompanied by stepwise changes not only in gene expression but also in cell morphology, mechanics and cell position. If any of these changes are instructive for differentiation itself, and whether consecutive changes in differentiation are required, remains unclear. Also, there are few comprehensive data sets on the exact changes in gene expression between different states of keratinocyte differentiation. In this study, through genetic fluorescence labeling of cell states at different developmental timepoints the authors were able to analyze gene expression of basal stem cells and suprabasal differentiated cells at two different stages of maturation: E14 (embryonic day 14) when the epidermis comprises mostly two functional compartments (basal stem cells and suprabasal so called intermediate cells) and E16 when the epidermis comprise three (living) compartments where the spinous layer separates basal stem cells from the barrier forming granular layer, as is the case in adult epidermis. Using RNA bulk sequencing, the authors developed useful new markers for suprabasal stages of differentiation like MafB and Cox1. The transcription factor MafB was then shown to inhibit suprabasal proliferation in a MafB transgenic model.

The data indicate that early in development at E14 the suprabasal intermediate cells resemble in terms of RNA expression, the barrier-forming granular layer at E16, suggesting that keratinocyte can undergo either stepwise (E16) or more direct (E14) terminal differentiation.

Previous studies by several groups found an increased actomyosin contractility in the barrier forming granular layer and showed that this increase in tension is important for epidermal barrier formation and function. However, it was not clear whether contractility itself serves as an instructive signal for differentiation. To address this question, the authors use a previously published model to induce premature hypercontractility in the spinous layer by using spastin overexpression (K10-Spastin) to disrupt microtubules (MT) thereby indirectly inducing actomyosin contractility. A second model activates myosin contractility more directly through overexpression of a constitutively active RhoA GEF (K10-Arhgef11CA). Both models induce late differentiation of suprabasal keratinocytes regardless of the suprabasal position in either spinous or granular layer indicating that increased contractility is key to induce late differentiation of granular cells. A potential weakness is the use of the K10-spastin model that disrupts MT and likely has additional roles in altering differentiation next to the induction of hypercontractility. Their previous publications provided some evidence that the effect on differentiation is driven by the increase in contractility (Ning et al. cell stem cell 2021). Moreover, their data are now further supported by a second model activating myosin through RhoA. This manuscript extends their previous findings that indicated a role for contractility in early differentiation, now focussing on the regulation of late differentiation in barrier forming cells. This data set thus help to unravel the interdependencies of cell position, mechanical state and differentiation in the epidermis, and suggest that an increase in cellular contractility within the epidermis can induce terminal differentiation. Importantly the authors show that despite contractility induced nuclear localization of the mechanoresponsive transcription factor YAP in the barrier forming granular layer, YAP nuclear localization is not sufficient to drive premature differentiation when forced to the nucleus in the spinous layer.

Overall, this is a well written manuscript and comprehensive dataset.

---

## [Referee Report · Reviewer #2 (Public review)]

Summary:

The manuscript from Prado-Mantilla and co-workers addresses mechanisms of embryonic epidermis development, focusing on the intermediate layer cells, a transient population of suprabasal cells that contributes to the expansion of the epidermis through proliferation. Using bulk-RNA they show that these cells are transcriptionally distinct from the suprabasal spinous cells and identify specific marker genes for these populations. They then use transgenesis to demonstrate that one of these selected spinous layer-specific markers, the transcription factor MafB is capable of suppressing proliferation in the intermediate layers, providing a potential explanation for the shift of suprabasal cells into a non-proliferative state during development. Further, lineage tracing experiments show that the intermediate cells become granular cells without a spinous layer intermediate. Finally, the authors show that the intermediate layer cells express high levels of contractility-related genes than spinous layers and overexpression of cytoskeletal regulators accelerates differentiation of spinous layer cells into granular cells.

Overall, the manuscript presents a number of interesting observations on the developmental stage-specific identities of suprabasal cells and their differentiation trajectories, and points to a potential role of contractility in promoting differentiation of suprabasal cells into granular cells. The precise mechanisms by which MafB suppresses proliferation, how the intermediate cells bypass the spinous layer stage to differentiate into granular cells and how contractility feeds into these mechanisms remain open. Interestingly, while the mechanosensitive transcription factor YAP appears differentially active in the two states, it is shown to be downstream rather than upstream of the observed differences in mechanics.

Strengths:

The authors use a nice combination of RNA sequencing, imaging, lineage tracing and transgenesis to address the suprabasal to granular layer transition. The imaging is convincing and the biological effects appear robust. The manuscript is clearly written and logical to follow.

Weaknesses:

While the data overall supports the authors claims, there are a few minor weaknesses that pertain to the aspect of the role of contractility, The choice of spastin overexpression to modulate contractility is not ideal as spastin has multiple roles in regulating microtubule dynamics and membrane transport which could also be potential mechanisms explaining some of the phenotypes. Use of Arghap11 overexpression mitigates this effect to some extent but overall it would have been more convincing to manipulate myosin activity directly. It would also be important to show that these manipulations increase the levels of F-actin and myosin II as shown for the intermediate layer. It would also be logical to address if further increasing contractility in the intermediate layer would enhance the differentiation of these cells.

Despite these minor weaknesses, the manuscript is overall of high quality, sheds new light on the fundamental mechanisms of epidermal stratification during embryogenesis and will likely be of interest to the skin research community.

---

## [Referee Report · Reviewer #3 (Public review)]

Summary:

This is an interesting paper by Lechler and colleagues describing the transcriptomic signature and fate of intermediate cells (ICs), a transient and poorly defined embryonic cell type in the skin. ICs are the first suprabasal cells in the stratifying skin and unlike later-developing suprabasal cells, ICs continue to divide. Using bulk RNA seq to compare ICs to spinous and granular transcriptomes, the authors find that IC-specific gene signatures include hallmarks of granular cells, such as genes involved in lipid metabolism and skin barrier function that are not expressed in spinous cells. ICs were assumed to differentiate into spinous cells, but lineage tracing convincingly shows ICs differentiate directly into granular cells without passing through a spinous intermediate. Rather, basal cells give rise to the first spinous cells. They further show that transcripts associated with contractility are also shared signatures of ICs and granular cells, and overexpression of two contractility inducers (Spastin and ArhGEF-CA) can induce granular and repress spinous gene expression. This contractility-induced granular gene expression does not appear to be mediated by the mechanosensitive transcription factor, Yap. The paper also identifies new markers that distinguish IC and spinous layers, and shows the spinous signature gene, MafB, is sufficient to repress proliferation when prematurely expressed in ICs.

Strengths:

Overall this is a well-executed study, and the data are clearly presented and the findings convincing. It provides an important contribution to the skin field by characterizing the features and fate of ICs, a much understudied cell type, at a high levels of spatial and transcriptomic detail. The conclusions challenge the assumption that ICs are spinous precursors through compelling lineage tracing data. The demonstration that differentiation can be induced by cell contractility is an intriguing finding, and adds a growing list of examples where cell mechanics influence gene expression and differentiation.

Weaknesses:

A weakness of the study is an over-reliance on overexpression and sufficiency experiments to test the contributions of MafB, Yap, and contractility in differentiation. The inclusion of loss-of-function approaches would enable one to determine if, for example, contractility is required for the transition of ICs to granular fate, and whether MafB is required for spinous fate. Second, whether the induction of contractility-associated genes is accompanied by measurable changes in the physical properties or mechanics of the IC and granular layers is not directly shown. Inclusion of physical measurements would bolster the conclusion that mechanics lies upstream of differentiation.

Finally, the role of ICs in epidermal development remains unclear. Although not essential to support the conclusions of this study, insights into the function of this transient cell layer would strengthen the overall impact.

---

## [Author Response]

The following is the authors’ response to the original reviews

**Reviewer #1 (Public review):**
Summary:The authors address a fundamental question for cell and tissue biology using the skin epidermis as a paradigm and ask how stratifying self-renewing epithelia induce differentiation and upward migration in basal dividing progenitor cells to generate suprabasal barrier-forming cells that are essential for a functional barrier formed by such an epithelium. The authors show for the first time that an increase in intracellular actomyosin contractility, a hallmark of barrier-forming keratinocytes, is sufficient to trigger terminal differentiation. Hence the data provide in vivo evidence of the more general interdependency of cell mechanics and differentiation. The data appear to be of high quality and the evidences are strengthened through a combination of different genetic mouse models, RNA sequencing, and immunofluorescence analysis.To generate and maintain the multilayered, barrier-forming epidermis, keratinocytes of the basal stem cell layer differentiate and move suprabasally accompanied by stepwise changes not only in gene expression but also in cell morphology, mechanics, and cell position. Whether any of these changes is instructive for differentiation itself and whether consecutive changes in differentiation are required remains unclear. Also, there are few comprehensive data sets on the exact changes in gene expression between different states of keratinocyte differentiation. In this study, through genetic fluorescence labeling of cell states at different developmental time points the authors were able to analyze gene expression of basal stem cells and suprabasal differentiated cells at two different stages of maturation: E14 (embryonic day 14) when the epidermis comprises mostly two functional compartments (basal stem cells and suprabasal socalled intermediate cells) and E16 when the epidermis comprise three (living) compartments where the spinous layer separates basal stem cells from the barrier-forming granular layer, as is the case in adult epidermis. Using RNA bulk sequencing, the authors developed useful new markers for suprabasal stages of differentiation like MafB and Cox1. The transcription factor MafB was then shown to inhibit suprabasal proliferation in a MafB transgenic model.The data indicate that early in development at E14 the suprabasal intermediate cells resemble in terms of RNA expression, the barrier-forming granular layer at E16, suggesting that keratinocytes can undergo either stepwise (E16) or more direct (E14) terminal differentiation.Previous studies by several groups found an increased actomyosin contractility in the barrierforming granular layer and showed that this increase in tension is important for epidermal barrier formation and function. However, it was not clear whether contractility itself serves as an instructive signal for differentiation. To address this question, the authors use a previously published model to induce premature hypercontractility in the spinous layer by using spastin overexpression (K10-Spastin) to disrupt microtubules (MT) thereby indirectly inducing actomyosin contractility. A second model activates myosin contractility more directly through overexpression of a constitutively active RhoA GEF (K10-Arhgef11CA). Both models induce late differentiation of suprabasal keratinocytes regardless of the suprabasal position in either spinous or granular layer indicating that increased contractility is key to induce late differentiation of granular cells. A potential weakness of the K10-spastin model is the disruption of MT as the primary effect which secondarily causes hypercontractility. However, their previous publications provided some evidence that the effect on differentiation is driven by the increase in contractility (Ning et al. cell stem cell 2021). Moreover, the data are confirmed by the second model directly activating myosin through RhoA. These previous publications already indicated a role for contractility in differentiation but were focused on early differentiation. The data in this manuscript focus on the regulation of late differentiation in barrier-forming cells. These important data help to unravel the interdependencies of cell position, mechanical state, and differentiation in the epidermis, suggesting that an increase in cellular contractility in most apical positions within the epidermis can induce terminal differentiation. Importantly the authors show that despite contractility-induced nuclear localization of the mechanoresponsive transcription factor YAP in the barrier-forming granular layer, YAP nuclear localization is not sufficient to drive premature differentiation when forced to the nucleus in the spinous layer.Overall, this is a well-written manuscript and a comprehensive dataset. Only the RNA sequencing result should be presented more transparently providing the full lists of regulated genes instead of presenting just the GO analysis and selected target genes so that this analysis can serve as a useful repository. The authors themselves have profited from and used published datasets of gene expression of the granular cells. Moreover, some of the previous data should be better discussed though. The authors state that forced suprabasal contractility in their mouse models induces the expression of some genes of the epidermal differentiation complex (EDC). However, in their previous publication, the authors showed that major classical EDC genes are actually not regulated like filaggrin and loricrin (Muroyama and Lechler eLife 2017). This should be discussed better and necessitates including the full list of regulated genes to show what exactly is regulated.

We thank the reviewers for their suggestions and comments.

Thank you for the suggestion to include gene lists. We had an excel document with all this data but neglected to upload it with the initial manuscript. This includes all the gene signatures for the different cell compartments across development. We also include a tab that lists all EDC genes and whether they were up-regulated in intermediate cells and cells in which contractility was induced. Further, we note that all the RNA-Seq datasets are available for use on GEO (GSE295753).

In our previous publication, we indeed included images showing that loricrin and filaggrin were both still expressed in the differentiated epidermis in the spastin mutant. Both Flg and Lor mRNA were up in the RNA-Seq (although only Flg was statistically significant), though we didn’t see a notable change in protein levels. It is unclear whether this is just difficult to see on top of the normal expression, or whether there are additional levels of regulation where mRNA levels are increased but protein isn’t. That said, our data clearly show that other genes associated with granular fate were increased in the contractile skin.

**Reviewer #2 (Public review):**
Summary:The manuscript from Prado-Mantilla and co-workers addresses mechanisms of embryonic epidermis development, focusing on the intermediate layer cells, a transient population of suprabasal cells that contributes to the expansion of the epidermis through proliferation. Using bulk-RNA they show that these cells are transcriptionally distinct from the suprabasal spinous cells and identify specific marker genes for these populations. They then use transgenesis to demonstrate that one of these selected spinous layer-specific markers, the transcription factor MafB is capable of suppressing proliferation in the intermediate layers, providing a potential explanation for the shift of suprabasal cells into a non-proliferative state during development. Further, lineage tracing experiments show that the intermediate cells become granular cells without a spinous layer intermediate. Finally, the authors show that the intermediate layer cells express higher levels of contractility-related genes than spinous layers and overexpression of cytoskeletal regulators accelerates the differentiation of spinous layer cells into granular cells.Overall the manuscript presents a number of interesting observations on the developmental stage-specific identities of suprabasal cells and their differentiation trajectories and points to a potential role of contractility in promoting differentiation of suprabasal cells into granular cells. The precise mechanisms by which MafB suppresses proliferation, how the intermediate cells bypass the spinous layer stage to differentiate into granular cells, and how contractility feeds into these mechanisms remain open. Interestingly, while the mechanosensitive transcription factor YAP appears deferentially active in the two states, it is shown to be downstream rather than upstream of the observed differences in mechanics.Strengths:The authors use a nice combination of RNA sequencing, imaging, lineage tracing, and transgenesis to address the suprabasal to granular layer transition. The imaging is convincing and the biological effects appear robust. The manuscript is clearly written and logical to follow.Weaknesses:While the data overall supports the authors' claims, there are a few minor weaknesses that pertain to the aspect of the role of contractility, The choice of spastin overexpression to modulate contractility is not ideal as spastin has multiple roles in regulating microtubule dynamics and membrane transport which could also be potential mechanisms explaining some of the phenotypes. Use of Arghap11 overexpression mitigates this effect to some extent but overall it would have been more convincing to manipulate myosin activity directly. It would also be important to show that these manipulations increase the levels of F-actin and myosin II as shown for the intermediate layer. It would also be logical to address if further increasing contractility in the intermediate layer would enhance the differentiation of these cells.

We agree with the reviewer that the development of additional tools to precisely control myosin activity will be of great use to the field. That said, our series of publications has clearly demonstrated that ablating microtubules results in increased contractility and that this phenocopies the effects of Arhgef11 induced contractility. Further, we showed that these phenotypes were rescued by myosin inhibition with blebbistatin. Our prior publications also showed a clear increase in junctional acto-myosin through expression of either spastin or Arhgef11, as well as increased staining for the tension sensitive epitope of alpha-catenin (alpha18). We are not aware of tools that allow direct manipulation of myosin activity that currently exist in mouse models.

The gene expression analyses are relatively superficial and rely heavily on GO term analyses which are of course informative but do not give the reader a good sense of what kind of genes and transcriptional programs are regulated. It would be useful to show volcano plots or heatmaps of actual gene expression changes as well as to perform additional analyses of for example gene set enrichment and/or transcription factor enrichment analyses to better describe the transcriptional programs

We have included an excel document that lists all the gene signatures. In addition, a volcano plot is included in the new Fig 2, Supplement 1. All our NGS data are deposited in GEO for others to perform these analyses. As the paper does not delve further into transcriptional regulation, we do not specifically present this information in the paper.

Claims of changes in cell division/proliferation changes are made exclusively by quantifying EdU incorporation. It would be useful to more directly look at mitosis. At minimum Y-axis labels should be changed from "% Dividing cells" to % EdU+ cells to more accurately represent findings

We changed the axis label to precisely match our analysis. We note that Figure 1, Supplement 1 also contains data on mitosis.

Despite these minor weaknesses the manuscript is overall of high quality, sheds new light on the fundamental mechanisms of epidermal stratification during embryogenesis, and will likely be of interest to the skin research community.
**Reviewer #3 (Public review):**
Summary:This is an interesting paper by Lechler and colleagues describing the transcriptomic signature and fate of intermediate cells (ICs), a transient and poorly defined embryonic cell type in the skin. ICs are the first suprabasal cells in the stratifying skin and unlike later-developing suprabasal cells, ICs continue to divide. Using bulk RNA seq to compare ICs to spinous and granular transcriptomes, the authors find that IC-specific gene signatures include hallmarks of granular cells, such as genes involved in lipid metabolism and skin barrier function that are not expressed in spinous cells. ICs were assumed to differentiate into spinous cells, but lineage tracing convincingly shows ICs differentiate directly into granular cells without passing through a spinous intermediate. Rather, basal cells give rise to the first spinous cells. They further show that transcripts associated with contractility are also shared signatures of ICs and granular cells, and overexpression of two contractility inducers (Spastin and ArhGEF-CA) can induce granular and repress spinous gene expression. This contractility-induced granular gene expression does not appear to be mediated by the mechanosensitive transcription factor, Yap. The paper also identifies new markers that distinguish IC and spinous layers and shows the spinous signature gene, MafB, is sufficient to repress proliferation when prematurely expressed in ICs.Strengths:Overall this is a well-executed study, and the data are clearly presented and the findings convincing. It provides an important contribution to the skin field by characterizing the features and fate of ICs, a much-understudied cell type, at high levels of spatial and transcriptomic detail. The conclusions challenge the assumption that ICs are spinous precursors through compelling lineage tracing data. The demonstration that differentiation can be induced by cell contractility is an intriguing finding and adds a growing list of examples where cell mechanics influence gene expression and differentiation.Weaknesses:A weakness of the study is an over-reliance on overexpression and sufficiency experiments to test the contributions of MafB, Yap, and contractility in differentiation. The inclusion of loss-offunction approaches would enable one to determine if, for example, contractility is required for the transition of ICs to granular fate, and whether MafB is required for spinous fate. Second, whether the induction of contractility-associated genes is accompanied by measurable changes in the physical properties or mechanics of the IC and granular layers is not directly shown. The inclusion of physical measurements would bolster the conclusion that mechanics lies upstream of differentiation.

We agree that loss of function studies would be useful. For MafB, these have been performed in cultured human keratinocytes, where loss of MafB and its ortholog cMaf results in a phenotype consistent with loss of spinous differentiation (Pajares-Lopez et al, 2015). Due to the complex genetics involved, generating these double mutant mice is beyond the scope of this study. Loss of function studies of myosin are also complicated by genetic redundancy of the non-muscle type II myosin genes, as well as the role for these myosins in cell division and in actin cross linking in addition to contractility. In addition, we have found that these myosins are quite stable in the embryonic intestine, with loss of protein delayed by several days from the induction of recombination. Therefore, elimination of myosins by embryonic day e14.5 with our current drivers is not likely possible. Generation of inducible inhibitors of contractility is therefore a valuable future goal.

Several recent papers have used AFM of skin sections to probe tissue stiffness. We have not attempted these studies and are unclear about the spatial resolution and whether, in the very thin epidermis at these stages, we could spatially resolve differences. That said, we previously assessed the macro-contractility of tissues in which myosin activity was induced and demonstrated that there was a significant increase in this over a tissue-wide scale (Ning et al, Cell Stem Cell, 2021).

Finally, whether the expression of granular-associated genes in ICs provides them with some sort of barrier function in the embryo is not addressed, so the role of ICs in epidermal development remains unclear. Although not essential to support the conclusions of this study, insights into the function of this transient cell layer would strengthen the overall impact.

By traditional dye penetration assays, there is no epidermal barrier at the time that intermediate cells exist. One interpretation of the data is that cells are beginning to express mRNAs (and in some cases, proteins) so that they are able to rapidly generate a barrier as they become granular cells. In addition, many EDC genes, important for keratinocyte cornification and barrier formation, are not upregulated in ICs at E14.5. We have attempted experiments to ablate intermediate cells with DTA expression - these resulted in inefficient and delayed death and thus did not yield strong conclusions about the role of intermediate cells. Our findings that transcriptional regulators of granular differentiation (such as Grhl3 and Hopx) are also present in intermediate cells, should allow future analysis of the effects of their ablation on the earliest stages of granular differentiation from intermediate cells. In fact, previous studies have shown that Grhl3 null mice have disrupted barrier function at embryonic stages (Ting et al, 2005), supporting the role of ICs in being important for barrier formation. (?)

**Recommendations for the authors:**

**Reviewer #1 (Recommendations for the authors):**
Overall, this is a well-written manuscript and a comprehensive dataset. Only the RNA sequencing result should be presented more transparently providing the full lists of regulated genes instead of presenting just the GO analysis and selected target genes so that this analysis can serve as a useful repository. The authors themselves have profited from and used the published dataset of gene expression of the granular cells. Moreover, some of the previous data should be better discussed though. The authors state that forced suprabasal contractility in their mouse models induces the expression of some genes of the epidermal differentiation complex (EDC). However, in their previous publication, the authors showed that major classical EDC genes are actually not regulated like filaggrin and loricrin (Muroyama and Lechler eLife 2017). This should be discussed better and necessitates including the full list of regulated genes to show what exactly is regulated.A general point regarding statistics throughout the manuscript. It seems like regular T-tests or ANOVAs have been used assuming Gaussian distribution for sample sizes below N=5 which is technically not correct. Instead, non-parametric tests like e.g. the Mann-Whitney test should be used. Since Graph-Pad was used for statistics according to the methods this is easy to change.Figure 1: It would be good to show the FACS plot of the analyzed and sorted population in the supplementary figures.If granular cells can be analyzed and detected by FACS, why were they not included in the RNA sequencing analysis?Figure 1 supplement 1c: cell division numbers are analyzed from only 2 mice and the combined 5 or 4 fields of view are used for statistics using a test assuming normal distribution which is not really appropriate. Means per mice should be used or if accumulated field of views are used, the number should be increased using more stringent tests. Otherwise, the p-values here clearly overstate the significance.

Granular cells could not be specifically isolated in the approach we used. The lectin binds to both upper spinous and granular cells. For this reason, we relied on a separate granular gene list as described.

For Figure 1 Supplement 1, we removed the statistical analysis and use it simply as a validation of the data in Figure 1.

Figure 2: It is not completely clear on which basis the candidate genes were picked. They are described to be the most enriched but how do they compare to the rest of the enriched genes. The full list of regulated genes should be provided.Some markers for IC or granular layer are verified either by RNA scope or immunofluorescence. Is there a technical reason for that? It would be good to compare protein levels for all markers. Figure 2-Supplement 1: There is no statement about the number of animals that these images are representative for.

We have included a volcano plot to show where the genes picked reside. We have also included the full gene lists for interested readers.

When validated antibodies were available, we used them. When they were not, we performed RNA-Scope to validate the RNA-Seq dataset.

We have included animal numbers in the revised Fig 2-Supplement 2 legend (previously Fig 2Supplement 1).

Figure 4b: It would be good to include the E16 spinous cells to get an idea of how much closer ICs are to the granular population.

We have included a new Venn diagram showing the overlap between each of the IC and spinous signatures with the granular cell signature in Fig 4B. Overall, 36% of IC signature genes are in common with granular cells, while just 20% of spinous genes overlap.

**Reviewer #2 (Recommendations for the authors):**
(1) Figure 6B is confusing as y-axis is labeled as EdU+ suprabasal cells whereas basal cells are also quantified.

We have altered the y-axis title to make it clearer.

(2) Not clear why HA-control is sometimes included and sometimes not.

We include the HA when it did not disrupt visualization of the loss of fluorescence. As it was uniform in most cases, we excluded it for clarity in some images. HA staining is now included in Fig 3C.

(3) The authors might reconsider the title as it currently is somewhat vague, to more precisely represent the content of the manuscript.

We thank the reviewer for the suggestion. We considered other options but felt that this gave an overview of the breadth of the paper.

**Reviewer #3 (Recommendations for the authors):**
(1) ICs are shown to express Tgm1 and Abca12, important for cornified envelope function and formation of lamellar bodies. Do ICs provide any barrier function at E14.5?

By traditional dye penetration assays, there is no epidermal barrier at the time that intermediate cells exist. One interpretation of the data is that cells are beginning to express mRNAs (and in some cases, proteins) so that they are able to rapidly generate a barrier as they become granular cells.

(2) Genes associated with contractility are upregulated in ICs and granular cells. And ICs have higher levels of F-actin, MyoIIA, alpha-18, and nuclear Yap. Does this correspond to a measurable difference in stiffness? Can you use AFM to compare to physical properties of ICs, spinous, and granular cells?

Several recent papers have used AFM of skin sections to probe tissue stiDness. We have not attempted these studies and are unclear about the spatial resolution and whether in the very thin epidermis at these stages whether we could spatially resolve diDerences. It is also important to note that this tissue rigidity is influenced by factors other than contractility. That said, we previously assessed the macro-contractility of tissues in which myosin activity was induced and demonstrated that there was a significant increase in this over a tissue-wide scale (Ning et al, Cell Stem Cell, 2021).

(3) Overexpression of two contractility inducers (spastin and ArhGEF-CA) can induce granular gene expression and repress spinous gene expression, suggesting differentiation lies downstream of contractility. Is contractility required for granular differentiation?

This is an important question and one that we hope to directly address in the future. Published studies have shown defects in tight junction formation and barrier function in myosin II mutants. However, a thorough characterization of differentiation was not performed.

(4) ICs are a transient cell type, and it would be important to know what is the consequence of the epidermis never developing this layer. Does it perform an important temporary structural/barrier role, or patterning information for the skin?

We have attempted experiments to ablate intermediate cells with DTA expression - this resulted in ineDicient and delayed death and thus did not yield strong conclusions. Our findings that transcriptional regulators of granular diDerentiation (such as Grhl3 and Hopx) are also present in intermediate cells, should allow future analysis of the eDects of their ablation on the earliest stages of granular diDerentiation from intermediate cells.